# Open Visual Knowledge Extraction via Relation-Oriented Multimodality Model Prompting

**Hejie Cui**[1][*]    **Xinyu Fang**[2][*]    **Zihan Zhang**[2]    **Ran Xu**[1]    **Xuan Kan**[1]    **Xin Liu**[3]
**Yue Yu**[4]    **Manling Li**[5]    **Yangqiu Song**[3]    **Carl Yang**[1][†]

[1]Emory University    [2]Tongji University    [3] The Hong Kong University of Science and Technology
[4] Georgia Institute of Technology    [5] Northwestern University

## Abstract

Images contain rich relational knowledge that can help machines understand the world. Existing methods on visual knowledge extraction often rely on the pre-defined format (e.g., sub-verb-obj tuples) or vocabulary (e.g., relation types), restricting the expressiveness of the extracted knowledge. In this work, we take a first exploration to a new paradigm of open visual knowledge extraction. To achieve this, we present `OpenVik` which consists of an open relational region detector to detect regions potentially containing relational knowledge and a visual knowledge generator that generates format-free knowledge by prompting the large multimodality model with the detected region of interest. We also explore two data enhancement techniques for diversifying the generated format-free visual knowledge. Extensive knowledge quality evaluations highlight the correctness and uniqueness of the extracted open visual knowledge by `OpenVik`. Moreover, integrating our extracted knowledge across various visual reasoning applications shows consistent improvements, indicating the real-world applicability of `OpenVik`.

## 1  Introduction

Knowledge extraction has been widely studied on texts [8, 1, 13, 9] for enhancing logical reasoning [45, 14, 6] and explainable AI [18, 57, 5, 55], and recent studies have explored *open* knowledge extraction through categorizing seed relations [64, 40] and eliciting from language models [47]. Visual knowledge extraction, on the other hand, captures intricate details like tools, sizes, and positional relationships, which are often difficult to express exhaustively in texts [39, 28, 48, 7]. Yet existing approaches of visual knowledge extraction are either restricted by a fixed knowledge format [52, 63, 20, 22] or the predefined sets of objects/relations [52, 63, 21]. While efficient at capturing interactions between objects, the produced visual knowledge is often limited in richness and confined to a single format, falling short in representing the diverse real-world information that can be complemented by visual data.

In this endeavor, we propose to further explore a new paradigm of open visual knowledge extraction (`OpenVik`). Specifically, we propose to generate relation-oriented, but format-free knowledge that includes a wider variety of elements, such as descriptions, insertions, and attributes, among others. Drawing inspiration from the wealth of knowledge encapsulated in large models [49, 61, 46], we propose to leverage pre-trained large multimodality models by eliciting open visual knowledge through relation-oriented visual prompting. This approach allows for a more nuanced understanding of visual data, mirroring how humans naturally emphasize certain aspects of visual scenes when perceiving and describing visual information, leading to more flexible visual knowledge extraction.

---

[*]These authors contributed equally to this work.
[†]Correspondence to: j.carlyang@emory.edu

37th Conference on Neural Information Processing Systems (NeurIPS 2023).

Our proposed `OpenVik` framework consists of two modules, an open relational region detector and a format-free visual knowledge generator. It is a unique challenge to detect the regions potentially containing relational knowledge, since traditional region detectors primarily focus on learning predefined object classes. To learn the regression of relational regions, we propose to use free-form knowledge descriptions as supervision and leverage knowledge generation as a training objective. With the detected regions, the remaining question is how to interpret these regions into free-form knowledge. We propose a visual knowledge generator by harnessing the power of language variety enhancement in large pre-trained multimodality models. Specifically, we prompt them to generate knowledge descriptions of any formats and condition the generation on the detected relational regions.

However, establishing a new paradigm of open visual knowledge extraction is challenging due to the absence of comprehensive and diverse training data. Existing datasets sources such as scene graphs [51, 24], dense captions [20], and dense relational subsets [22] often exhibit a long-tail distribution biased to more prevalent relations and entities [44]. Brute-force merging of these datasets could exacerbate the distribution bias inherent in the data. To alleviate the bias, we propose two diversity-driven data enhancement strategies based on an adapted TF-IDF+ score, involving random dropping and data augmentation with external knowledge resources. These strategies optimize data distributions and richness, thus fostering diverse open visual knowledge extraction.

We implement extensive evaluations to assess the quality and utility of the open visual knowledge extracted by `OpenVik`, encompassing: 1) directly evaluating the performance of knowledge generation; 2) engaging human evaluators for a multi-faceted assessment of in-depth knowledge quality; and 3) comparing the open visual knowledge extracted with `OpenVik` with existing knowledge sources, such as non-parametric knowledge from the ConceptNet knowledge graph, and parametric knowledge from the GPT-3.5 large language model. Furthermore, the utility of the extracted open visual knowledge is validated through its integration with several common applications that require visual understanding, including text-to-image retrieval, grounded situation recognition, and visual commonsense reasoning. These applications demonstrate consistent improvements, affirming the practical utility of `OpenVik`.

## 2   Related Work

**Visual knowledge extraction.** Recent advancements in knowledge extraction have extended from being purely text-driven to incorporating images [11, 29]. VisKE [39] is designed to verify relations between pairs of entities, e.g., *eat*(*horse*, *hay*). Scene graphs, which locate objects in the image and identify visual predicates between subjects and objects in a triple format, e.g., (*man, on, chair*), are extensively studied for vision understanding [52, 63, 60]. A recent work OpenSGG [17] extends SGG to open-vocabulary objects, enabling the relation prediction for unseen objects. Other studies have explored caption-like formats, like dense captioning [20] with a set of object-centric descriptions across regions, and relational captioning [22] focusing on relational information between objects. Despite these advancements, existing methods either adhere to a pre-defined format and vocabulary or are constrained by the biased distribution of training sets. This highlights the pressing need for a format-free approach in visual knowledge extraction with knowledge diversity.

**Large model prompting.** Recently, large language and multimodality models have exhibited remarkable successes in capturing commonsense knowledge across various tasks, especially facilitating few-shot [15, 53, 25, 58] and zero-shot learning [23, 66, 59]. The potential of prompt-based learning for pre-trained vision-language models [2, 37, 42] has been explored for handling diverse data types across multiple modalities, such as images and texts, with improved performance in tasks including image classification [33, 67], segmentation [32] and visual question answering [16]. Leveraging the substantial information encapsulated within these pre-trained multimodality models to extract explicit knowledge can enrich existing resources, potentially laying the groundwork for advances in interpretability research and mitigating the hallucination issue associated with large models [19, 10].

## 3   Method

In this section, we introduce our new paradigm and two key model design novelty featuring `OpenVik`, relation-oriented multimodality model prompting and diversity-driven data enhancement.

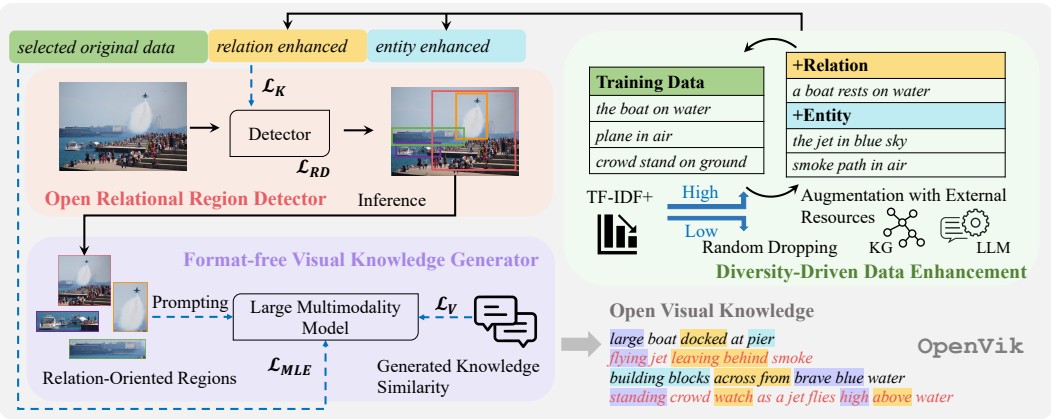

Figure 1: The overview of `OpenVik`. The left orange and purple panels illustrate key components of relation-oriented multimodality model prompting: open relational region detector and format-free visual knowledge generator. The right green one depicts diversity-driven data enhancement strategy. `OpenVik` is designed to extract relation-oriented format-free open visual knowledge with novel entities, diverse relations, and nuanced descriptive details.

## 3.1 Open Visual Knowledge Extraction

Given a dataset $\mathcal{D} = \{(\mathcal{I}_i, \mathbf{T}_i, \mathbf{U}_i)\}_{i=1}^M$ consisting of $M$ samples, $\mathcal{I}_i$ is the $i$-th image (such as the input image in Figure 1), $\mathbf{T}_i = \{\mathcal{T}_j\}_{j=1}^{n_i}$ is a set of $n_i$ region descriptions (such as "*the boat on water*" in Figure 1), $\mathbf{U}_i = \{\mathcal{U}_j\}_{j=1}^{n_i}$ is the set of $n_i$ relation-oriented visual regions, where each $\mathcal{T}_j$ corresponds to a visual region $\mathcal{U}_j \in \mathbf{U}_i$ in image $\mathcal{I}_i$. The goal of our open visual knowledge discovery is to train a model $\mathcal{M}$ capable of producing a set of format-free knowledge descriptions (such as "*large boat docked at pier*" in Figure 1) given any image $\mathcal{I}_k$ during the inference stage.

## 3.2 Relation-Oriented Multimodality Model Prompting

The overall architecture of `OpenVik` is shown in Figure 1. It comprises two modules: an open relational region detector $\mathcal{M}_v$ and a format-free visual knowledge generator $\mathcal{M}_t$. The two modules are learned separately during training with our diversity-enhanced data (Section 3.3) and combined to produce format-free visual knowledge at inference. Specifically, the relational region detector $\mathcal{M}_v$ takes an image $\mathcal{I}_i$ as the input and learns to select a flexible number of relational regions $\mathbf{U}_i = \{(\mathcal{U}_j)\}_{j=1}^{n_i}$ that captures object interactions, each corresponding to a description $\mathcal{T}_j$ in $\mathbf{T}_i$; the visual knowledge generator $\mathcal{M}_t$ generates format-free knowledge descriptions by prompting and fine-tuning the multimodality model with the guidance of detected visual region $\mathcal{U}_j$. All notations for the region detector and knowledge generator are detailed in Table 9 and Table 10, respectively.

Table 1: Notations for open region detector.

| Notation | Meaning |
|---|---|
| $\mathcal{I}_i$ | input image of the relational region detector |
| $\mathcal{U}_j$ | relation-centric box label |
| $\mathbf{U}_i$ | set of relation-centric boxes of an image |
| $\mathcal{T}_j$ | region description of a box |
| $\mathbf{T}_i$ | set of region descriptions of the an image |
| $\mathcal{L}_{RD}$ | region regression loss supervised by union regional boxes |
| $\mathcal{L}_K$ | knowledge generation loss supervised by GT relational knowledge |
| $\mathcal{L}_v$ | the overall objective of the relational region detector |

Table 2: Notations for knowledge generator.

| Notation | Meaning |
|---|---|
| $\mathcal{I}_i$ | the input image of the knowledge generator |
| $T_a, T_b$ | two regional knowledge descriptions of one same image |
| $N_i$ | the number of generated knowledge descriptions of an image |
| $\phi$ | hyper-parameter controlling the penalty slightly different sequences |
| $\mathcal{L}_{MLE}$ | the language modeling loss of the generation decoder |
| $\mathcal{L}_V$ | inter-sequence information variety regularizer |
| $\alpha$ | weight hyper-parameter balancing generation accuracy and variety |
| $\mathcal{L}_l$ | the overall objective of the knowledge generator |

**Open relational region detector.** Although existing object detection algorithms have been widely recognized for their efficiency in object detection, they are usually restricted to object-centric visual regions in a predefined set, and thus cannot directly capture open relational information with a single box. Detecting regions containing relational knowledge remains to be a challenge. We make two adaptions on the object detection FasterRCNN [38] to train the open relational region detector:

- *Region Regression*: we change the original object-centric region labels to our newly created relation-centric box labels, denoted as $\mathbf{U}_j$. The foreground of each relation-centric region label $\mathcal{U}_j$ is created by taking the union of the object-level bounding boxes of the entities, i.e., *boat, water*, contained in a ground truth region knowledge description $\mathcal{T}_j$. This forms the region regression loss $\mathcal{L}_{RD}$.

- *Knowledge Supervision*: To assist with the refinement of the bounding box, we replaced the object-centric label classification in traditional object detectors with knowledge supervision. A pre-trained generator is finetuned to create the regional description grounded to the given region. This is supervised by the cross-entropy loss $\mathcal{L}_K$ with region description $\mathbf{T}_j$.

The training objective $\mathcal{L}_l$ of the relational region detector is formulated as below, where $\mathcal{L}_{RD}$ is the regional regression loss and $\mathcal{L}_K$ is the knowledge supervision loss,

$$\mathcal{L}_v = \mathcal{L}_{RD} + \mathcal{L}_K. \tag{1}$$

**Format-free visual knowledge generator.** OpenVik provides better knowledge grounding by conditioning the generator on the detected relational region, leading to a reasoning-driven generation. Specifically, the detected bounding box (such as the box containing "*boat*" and "*pier*" on the far left) is utilized as a visual prompt when fine-tuning the visual knowledge generator. The model architecture of the knowledge generator is built upon a combined large multimodality model, which composes a pre-trained vision transformer ViT-B [12] and the image-grounded text decoder of BLIP [27]. The two modules are jointly trained on a generic image-text paired dataset comprising over 14 million entries and fine-tuned on the image captioning task, which delivered state-of-the-art performance.

In our visual knowledge generator, the decoder takes the ViT visual representation of the entire image as input and leverages the detected regional mask as a binary visual prompt. This prompt aids in filtering out the background and directing attention toward the relational foreground. The generation of format-free knowledge from the decoder is supervised by the language modeling loss $\mathcal{L}_{MLE}$, which further refines visual attention during the knowledge generation process. As a result, our approach facilitates the production of format-free outcomes that extend beyond the conventional sub-verb-obj form. Besides, to improve information variety, we introduce an amplifying penalty factor for highly similar knowledge generation. For any two generated sequences $T_a$ and $T_b$ describing image $\mathcal{I}_i$,

$$\mathcal{L}_V = \frac{1}{N_i} \sum_{N_i} \text{ReLU}\left(-\log\left(1 - \left(s\left(T_a, T_b\right) - \phi\right)\right)\right), \tag{2}$$

where $N_i$ is the number of generated knowledge of image $\mathcal{I}_i$, $s\left(T_a, T_b\right)$ indicates the semantic cosine similarity, and $\phi$ is a hyper-parameter set as 0.01 controlling the penalty on sequences with only slight difference (e.g. "*dog chasing the man*" and "*dog licking the man*") to be relatively small.

The training objective $\mathcal{L}_l$ of the format-free visual knowledge generator is formulated as

$$\mathcal{L}_l = \alpha \times \mathcal{L}_{MLE} + (1 - \alpha) \times \mathcal{L}_V, \tag{3}$$

where $\alpha$ is a weighting hyper-parameter we set as 0.7. The trained relational region detector and visual knowledge generator are combined during inference. Given any image $\mathcal{I}$, the open relational region detector first detects a flexible number of open relations regions of interest, then each detected region $\mathcal{R}$ is passed to the format-free visual knowledge generator, where a relation-oriented format-free knowledge phrase (such as "*flying jet leaving behind smoke*" in Figure 1) is generated to describe the given visual focus subarea $\mathcal{R}$ of the image. To further encourage within-sequence language variety during inference, we leverage the contrastive decoding strategy from [43], which improves over nucleus sampling and beam search.

### 3.3 Diversity-driven Data Enhancement

The training data for relational knowledge extraction usually exhibits a long-tail distribution, where more prevalent but simple relations such as *in*, *on*, and *wear* dominate the training set [44]. Consequently, the model trained with such a biased dataset may render limited, and repetitive knowledge. As a remedy, we propose two data enhancement techniques to optimize the data distribution. As the foundational measure for given relation $r$'s importance, we design a grid TF-IDF+ score $\mathcal{S}_r$ [34, 54]:

$$\mathcal{S}_r = (\log(\frac{N}{1 + f_r * \alpha_1}))^{\alpha_2}, \tag{4}$$

where $N$ is the total number of knowledge phrases in the datasets, $f_r$ is the number of occurrences of the relation $r$, $\alpha_1$ and $\alpha_2$ are the grid scales whose values are selected based on $f_r$.

**Random dropping on low-quality data.** We first remove repeated knowledge descriptions in the same image and then randomly drop descriptions that contain frequently occurring yet meaningless

relations with a low $\mathcal{S}_r$ (e.g., "*people on ground*") from the original dataset. Specifically, if the $\mathcal{S}_r$ of the relation in a description is relatively low, i.e., 0.4, we remove it at a random dropping rate of 0.5. This process repeats for all descriptions in an image until the remaining set is 0.6 times the size of the original training set. Consequently, the training data bias is mitigated by removing low-quality data.

**Data augmentation with external knowledge resources.** For the relations with high TF-IDF+ scores, we leverage external knowledge resources from both non-parametric (i.e., ConceptNet [41]) and parametric (i.e., COMET [4]) knowledge resources to promote diverse knowledge generation [56]. ✔ *Enhance Relation Recognition*: For each training description with a high TF-IDF+ score, we perform semantic parsing to get all the objects and complement additional relations (e.g., "*rest*" in Figure 1) between each pair of them by mapping the nodes and retrieving edges from the ConceptNet. Each retrieved knowledge triplet is converted to a knowledge phrase and added to the training set for generator training. With this introduced external knowledge, the knowledge generator ultimately yields a more robust and detailed representation of the underlying visual information of objects. This, in turn, bolsters the relation recognition of the visual knowledge generator. ✔ *Boost Entity Perception*: For the description with the highest-scored TF-IDF+ relation given each image, we also leverage ConceptNet to enrich similar objects (e.g., "*jet*") to the original object (e.g., "*plane*"). Additionally, we further introduce new entities (e.g., "*smoke*" in Figure 1) and attribute descriptions (e.g., "*blue*") by prompting the pre-trained attribute commonsense branch of the COMET model (Refer to Appendix A for more details). The entity-based enrichment potentially helps in boosting entity understanding and at the same time enhances the occurrence of important but rare relations in the training set.

## 3.4 Implementation Details

Our training data are built based on Visual Genome [24] and its relation-enhanced version Dense Relational Captioning [22]. Each sample includes an image identified by a unique ID and a set of relational descriptors describing interactions among objects in the image. Specifically, each relational descriptor includes the full description text, the subject and object names contained in the description text, the relation between them, as well as the bounding box coordinates of the subject and object. The dataset statistic information is summarized in Table 8 in the Appendix B.

Our model is implemented in PyTorch [35] and trained on two Quadro RTX 8000 GPUs. The open relational region detector is initialized from the ResNet50-FPN backbone, then finetuned for another 20 epochs with the relational bounding box. The model detects a maximum of 30 bounding boxes for each image with the highest confidence to avoid misleading noises. The format-free visual knowledge generator is initialized from BLIP$_{base}$ with the basic ViT-B/16 and finetuned for 20 epochs. Full details on learning parameters can be referred to in Appendix C.

# 4 Evaluation

In this section, we directly evaluate the extracted open visual knowledge from `OpenVik` from two perspectives: (1) knowledge generation performance with traditional generative metrics and in-depth knowledge quality assessment; (2) comparison with existing knowledge sources. Besides, ablation studies are conducted to study the influence of diversity design on the generated knowledge and data.

## 4.1 Evaluation on Generated Knowledge

**Generation performance.** To directly evaluate the visual knowledge generator, we compare the knowledge generated by `OpenVik` with a variety of baselines, including scene graph generation [52, 63, 44, 17] (of which Ov-SGG employs an open vocabulary), dense relational captioning [22], and region captioning [20, 65, 27, 26]. Evaluation metrics are traditional language generation measures such as BLEU, ROUGE-L, and METEOR. The results, displayed in the left side of Table 3, reveal that `OpenVik` outperforms captioning-based approaches and yields results on par with the best scene graph generation baseline. These findings underscore the effectiveness of the format-free visual knowledge generator through relation-oriented prompting of the large multimodality model.

**In-depth knowledge quality.** To more thoroughly evaluate the quality and richness of the format-free visual knowledge extraction, beyond simply evaluating it as a language generation model with the

Table 3: Knowledge comparison of `OpenVik` and baselines on performance and in-depth quality (%).

| Method | Generation Performance | | | In-Depth Knowledge Quality | | | |
|---|---|---|---|---|---|---|---|
| | BLEU↑ | ROUGE-L↑ | METEOR↑ | Validity↑ | Conformity↑ | Freshness↑ | Diversity↑ |
| *Closed/Open Scene Graph Generation* | | | | | | | |
| IMP [52] | 0.075 | 0.123 | 0.118 | 0.800 | 0.823 | 0.676 | 0.316 |
| Neural Motifs [63] | 0.229 | 0.283 | **0.273** | 0.822 | 0.767 | 0.667 | 0.349 |
| UnbiasSGG [44] | 0.217 | 0.258 | 0.194 | 0.739 | 0.733 | 0.666 | 0.357 |
| Ov-SGG [17] | 0.167 | 0.210 | 0.183 | 0.712 | 0.633 | 0.693 | 0.413 |
| *Dense Relational Captioning* | | | | | | | |
| MTTSNet+REM [22] | 0.240 | 0.226 | 0.228 | 0.897 | 0.852 | 0.754 | 0.375 |
| *Region Captioning* | | | | | | | |
| DenseCap [20] | 0.248 | 0.245 | 0.196 | 0.883 | 0.843 | 0.790 | 0.543 |
| Sub-GC [65] | 0.272 | 0.263 | 0.221 | 0.892 | 0.871 | 0.795 | 0.547 |
| BLIP [27] | 0.264 | 0.266 | 0.252 | 0.886 | 0.855 | 0.760 | 0.531 |
| BLIP2 [26] | 0.275 | **0.285** | 0.257 | 0.892 | 0.871 | 0.766 | 0.535 |
| *Open Visual Knowledge Extraction* | | | | | | | |
| `OpenVik` | **0.280** | 0.283 | 0.250 | **0.907** | **0.883** | **0.809** | **0.619** |

limitation of training data, we incorporate four additional metrics [31], which delve into an in-depth quality evaluation of the extracted visual knowledge from four distinct perspectives:

- *Validity* (↑): whether the generated visual knowledge is valid to human.
- *Conformity* (↑): whether the generated knowledge faithfully depicts the scenarios in the images.
- *Freshness* (↑): the novelty of the knowledge, i.e., the proportion not present in the training set.
- *Diversity* (↑): the language variance between a randomly sampled pair of knowledge pieces.

Among the four metrics, both the validity and conformity metrics involve human annotators. We randomly selected 100 images as the evaluative subset. Details regarding the scoring guidance and the interface provided to the annotators can be found in Appendix D. The remaining metrics, i.e., freshness and diversity, are calculated automatically. The in-depth knowledge quality evaluation results are displayed in the right part of Table 3, where the average pairwise Cohen's $\kappa$ on human evaluation results is 0.76 (good agreement). The findings demonstrate that trained with the diversity-enhanced datasets, the format-free visual knowledge extracted by `OpenVik` significantly outperforms other types of baselines in terms of all four metrics. The improvement of diversity, in particular, reaches 14% relatively compared with the inference results from the second runner DenseCap, indicating the advantage of `OpenVik` in generating rich and comprehensive visual knowledge.

## 4.2 Comparison with Existing Knowledge Sources

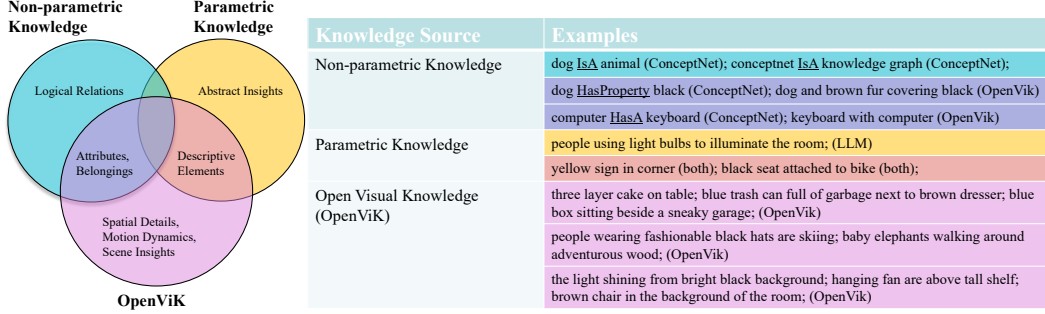

Figure 2: The Venn diagram of knowledge comparison between the open visual knowledge from `OpenVik` with the non-parametric knowledge from existing knowledge graph (i.e., ConceptNet) and parametric knowledge from large language model (i.e., COMET).

We compare the extracted visual knowledge with the non-parametric knowledge in the existing knowledge graph (KG) and the parametric knowledge from the large language model (LLM). The comparison insights from the three knowledge resources are shown in the Venn Diagram in Figure 2.

**Compare with non-parametric knowledge.** We take ConceptNet [41] as the representative in the comparison with non-parametric knowledge. To map the knowledge generated by `OpenVik` to ConceptNet, we parse the knowledge into triplets and associate the endpoints of these triplets with

nodes in ConceptNet. Then we calculate the similarity of embeddings[3] between the parsed relation and all the edge relations among the mapped nodes in ConceptNet. If the similarity score exceeds a predetermined threshold, i.e., 0.75, we consider the mapping successful. As illustrated in Figure 2, we observe that compared with the non-parametric knowledge in KG, the extracted visual knowledge captures richer and more meaningful spatial details, e.g., "*three layer cake on table*", and motion dynamics, e.g., "*baby elephants walking around adventurous wood*".

**Compare with parametric knowledge.** We compare with parametric knowledge contained in LLM by prompting the gpt-3.5-turbo[4] model with the object information in the image. The prompt template used is detailed in Appendix E. The mapping process follows the approach mentioned earlier. It is found that compared with the parametric knowledge in LLM, the extracted visual knowledge exhibits unique fine-grained visual details, e.g., "*red sticker on fence*", and provides precise scene information, e.g., "*the light shining from bright black background*".

## 4.3 Ablation Study

**The influence on knowledge quality with information variety regularization and data strategies.** We conducted ablation studies to evaluate the effectiveness of the information variety regularizer, $\mathcal{L}_V$, and our diversity-driven data enhancement strategies. This involves an in-depth assessment of knowledge quality on the same eval-

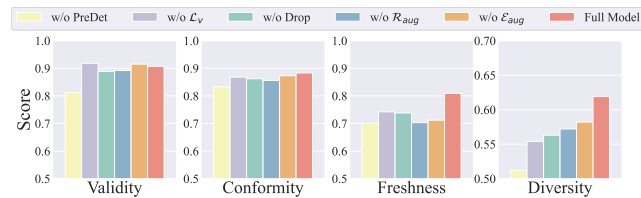

Figure 3: The influence of information variety regularization and diversity-driven data enhancement strategies.

uation subset. The results are presented in Figure 3. It is evident from the results that our proposed information variety design primarily impacts freshness and diversity, without compromising validity and conformity. For the freshness, the omission of data augmentation for entities and relations results in the most significant performance degradation. This implies the crucial role these strategies play in infusing novel knowledge into the generation process. As for diversity, the most notable changes in metrics are observed when the $\mathcal{L}_V$ and random dropping are removed. The strategy for augmenting entities and relations also plays a valuable role in enriching diversity.

**Ablation of the pre-training for the open relational region detector.** We conducted a comparison of the outcomes when loading a pre-trained detector backbone versus training the detector from scratch, as shown by the yellow bar in Figure 3. Results demonstrate a noticeable decrease in both knowledge diversity and freshness, which indicates the importance of loading the pre-trained model for region detection. This may be because omitting the pre-training step of the FasterRCNN model tends to result in the detection of more overlapping regions, which in turn causes the drop.

**The influence on dataset diversity with data strategies.** We conduct a direct analysis of the knowledge diversity of the existing datasets and our diversity-enhanced one, compared with the visual knowledge generated from OpenVik. The findings, presented in Table 4, show that the diversity-driven data enhancement strategies significantly boost knowledge diversity. Trained with this enhanced data, OpenVik can extract visual knowledge that exhibits greater diversity than that found in the *Visual Genome* and *Relational Caps*, indicating the advantage of OpenVik to format-free visual knowledge generation and its ability to yield richer knowledge diversity.

Table 4: Diversity of existing and enhanced datasets and generated knowledge from OpenVik.

| Metrics | Training Dataset | | | Generate Knowledge |
| --- | --- | --- | --- | --- |
| | *Visual Genome* [24] | *Relational Caps* [22] | *Diversity Enhanced (Ours)* | OpenVik *(Ours)* |
| **Diversity** | 0.589 | 0.604 | 0.632 | 0.619 |

## 4.4 Case Study

We present two case studies in Figure 4 (See Appendix F for more) to showcase the format-free visual knowledge generated by OpenVik, in comparison to Visual Genome (Scene Graph and Region Description) and Relational Caps. Contrary to the rigidity of scene graphs, which strictly adhere to a

---

[3]Embeddings are produced by ConceptNet API: https://github.com/commonsense/conceptnet-numberbatch.
[4]https://platform.openai.com/docs/models/gpt-3-5

predefined format, `OpenVik` can generate knowledge with a flexible semantic structure, not strictly bound to the sub-verb-obj format (e.g., "*blue post attached to wall with white letter*"). Examples of this adaptability are highlighted in red. When compared to dense region descriptions, the relational knowledge extracted by `OpenVik` offers a deeper understanding of the multiple entity interactions within an image. In comparison to Relational Caps, which mainly focus on interactions between two objects, `OpenVik` significantly broadens the diversity of relation with vivid verbs (e.g., "*attached to*", "*adorning*"). Moreover, it introduces novel entities (e.g., "*post*", "*mane*") and enriches the knowledge representation with nuanced details (e.g., "*full of*", "*striped*") that are missed by Relational Caps.

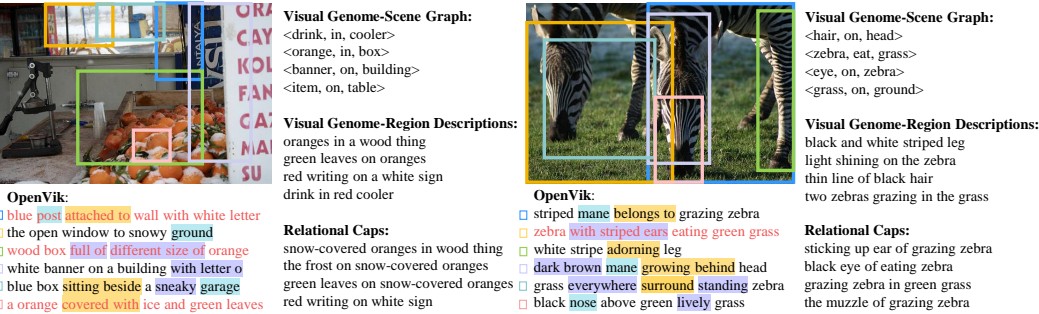

Figure 4: Case study on the extracted open visual knowledge from `OpenVik`. Examples of format-free knowledge are highlighted in red. Compared with VG and Relational Caps, `OpenVik` performs better at capturing novel entities , broadening object interactions with diverse relations , and enriching the knowledge representation with nuanced descriptive details .

Note that we observe the unbalanced and noisy distributions within the training data can lead to errors in the knowledge produced. Viewing hallucinations as erroneous inferences based on input, the inaccuracies observed in OpenVik and similar baselines often stem from detection errors. These errors are typically caused by data biases that incorrectly associate features with a specific class or label. We further two illustrative failure cases in Figure 5. For example, a "*black speaker by flat tv*" is generated, although the speaker is not present in the image—possibly reflecting common co-occurrences within the dataset. Similarly, a ladder in the right figure has been misidentified as a towel, leading to the erroneous description of a "*blue towel hanging from dry shower*". The key to mitigating such incorrect inference is identifying the cofounder feature of class labeling.

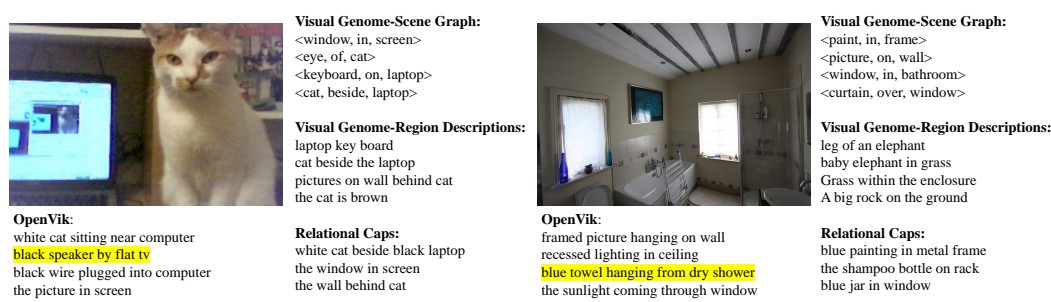

Figure 5: Examples of incorrectly knowledge resulting from distribution bias are highlighted .

## 5 Application

This section explores whether the extracted open visual knowledge from `OpenVik` can bolster reasoning and inference capabilities in multimodality downstream tasks by augmenting a baseline in the challenging zero-shot setting.

### 5.1 Text-to-Image Retrieval

**Task Setting.** In the text-to-image retrieval task, a given caption is matched to a large set of candidate images, with the most relevant image returned as the result. Adopting the challenging zero-shot

setting, we generate the visual representation $v$ and textual representation $t$ of the given image $\mathcal{I}$ and caption $\mathcal{T}$ using a pre-trained clip-retrieval model [3]. The baseline involves the image and text embedding similarly based on zero-shot CLIP Retrieval [3] and the fine-tuned model from BLIP [26].

To explore the potential of the extracted visual knowledge from OpenVik, we enrich the given caption $\mathcal{T}$ with related contexts derived from the extracted visual knowledge. Specifically, for each query caption, we parse the caption to extract all subject-object pairs $(s, o)$ with the NLTK parser. Then $s$ and $o$ are mapped to the open visual knowledge, where knowledge phrases that contain relations occurring more than 30% of the time between $s$ and $o$ are enriched to the original caption $\mathcal{T}$.

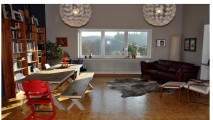

**Original text:** A little room and dining room area with furniture. A living room with a big table next to a book shelf. A living room decorated with a modern theme. A living room with wooden floors and furniture. The large room has a wooden table with chairs and a couch.

**Enriched text:** big table in room. a decorated living room with wooden furniture. brown couch in room. book on table. wooden table with shelf. shelf next to couch. wooden bookshelf with books next to table.

Figure 6: An example of OpenVik enrichment on text-to-image retrieval (See Appendix G.1 for more).

| Method | Recall@1 | Recall@5 | Recall@10 | Avg |
|---|---|---|---|---|
| ZS-CLIP | 36.16 | 65.47 | 78.66 | 60.10 |
| OpenVik + ZS-CLIP | 40.55 | 73.29 | 84.53 | 66.12 |
| BLIP | 63.11 | 86.30 | 91.10 | 80.17 |
| OpenVik + BLIP | 65.23 | 87.71 | 91.90 | 81.61 |

Table 5: Text-to-image retrieval results (%) of OpenVik enrichment compared with zero-shot baselines.

**Qualitative examples.** Figure 6 presents an example of OpenVik-based visual knowledge enrichment on captions. By incorporating related contexts from the generated open visual knowledge, the enriched captions convey more precise visual details, which enhances the alignment for text-image alignment.

**Quantitative results.** We curated a subset of 680 images from the testing set of the MS-COCO dataset containing parsed knowledge with at least eight nouns. This ensures an adequate degree of enrichment is achieved through the use of OpenVik. Standard image retrieval metrics, i.e., *Recall@1/5/10/* and *Avg*, are employed to evaluate the performance. The results are presented in Table 5. It is evident that relational context enrichment leads to the average correction of more than 6.0% of the initial zero-shot, highlighting the practical benefits of extracted visual knowledge in visual reasoning tasks.

## 5.2 Grounded Situation Recognition

**Task setting.** The event type prediction for the grounded situation recognition task is to predict the best match from predefined 504 event types [36] based on the image. We convert each candidate event verb into a description $\mathcal{T}$: "*An image of <verb>*" for image description matching. Similarly to text-to-image retrieval, we include zero-shot CLIP and the fine-tuned model from BLIP as baselines.

To enrich with contextual knowledge from OpenVik, for each given verb $v$, we find its nearest synonym in the extracted open visual knowledge and enrich the text description with the most common knowledge phrase containing it, regularized by the objects present in the image. Instead of directly concatenating the retrieved knowledge triplets to the original textual description, we employ an additive decomposition strategy: the similarity $s(\mathcal{I}, v)$ of the candidate verb $v$ with respect to the given image $\mathcal{I}$ is calculated as $s(\mathcal{I}, v) = \frac{1}{|D(v)|} \sum_{d \in D(v)} \phi(\mathcal{I}, v)$, where $D(v)$ is the set of descriptors, including the original description and the enriched ones, and $\phi$ represents the single $\log$ probability that descriptor $d$ pertains to the image $\mathcal{I}$.

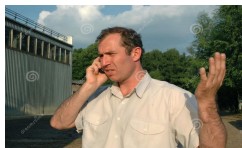

**Verb:** talking
**Original text:** This is an image of talking.
**Enriched text:** man talking on a small white telephone. adult male with white shirt talking on chatty cellphone. the man waving arms is talking on phone.

Figure 7: An example of OpenVik context enrichment on task GSR (See Appendix G.2 for more).

| Method | Accuracy | Precision | Recall | F$_1$ |
|---|---|---|---|---|
| ZS-CLIP | 53.14 | 42.54 | 45.19 | 43.82 |
| OpenVik + ZS-CLIP | 75.16 | 61.63 | 62.75 | 62.18 |
| BLIP | 70.42 | 65.32 | 69.25 | 67.23 |
| OpenVik + BLIP | 80.25 | 72.55 | 70.61 | 71.57 |

Table 6: Grounded situation recognition results (%) of OpenVik enrichment compared with zero-shot baselines.

**Qualitative examples.** Figure 7 presents a qualitative example of OpenVik-based context enrichment in the grounded situation recognition task. We observed that verbs like "shopping" and "talking" were appropriately enriched with their frequently occurring contexts from the open visual knowledge, leading to a reduced embedding distance between the description and its matching image.

**Quantitative results.** We assembled a test set of 900 samples from the testing set of GSR that included verbs such as "talking", "filming", and "picking", among others, from a list of 256 words

that can be accurately mapped to extracted visual knowledge, as well as 138 verbs that have a fuzzy match through ConceptNet embedding comparison. The full lists of the exact and fuzzy-matched verbs are detailed in Appendix H. The evaluated metrics include Accuracy, Precision, Recall, and $F_1$. The results are presented in Table 6. It can be observed that knowledge enrichment significantly outperforms the zero-shot and BLIP baselines. This suggests that the verb-related contexts introduced by `OpenVik`-generated knowledge are intuitive and greatly assist in understanding the semantics of event verbs, bolstered by related visual information.

### 5.3 Visual Commonsense Reasoning

**Task setting.** The goal of visual commonsense reasoning is to predict an answer from four given option candidates for a given image and question. For the baseline approach, we compare the backbone model R2C from the VCR paper [62] and BLIP [27]. In the visual knowledge-enhanced `OpenVik` Enriched approach, we perform two-level context augmentation, incorporating both entities and relations: (1) we parse the question and options to obtain all (S, O) pairs and, for each entity pair, apply the same relation augmentation as in the image retrieval task; (2) for the V in each option, we enrich the visual context using the same method as illustrated in grounded situation recognition.

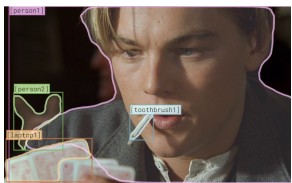

**Question** : Is Person1 winning the game? the person engaged in a game.
**A** Yes, he is about to go for a run. the person engaged in a game. the person walking near runway.
**B** No, he is losing. the person engaged in a game. frustrated person lose game.
**C** No, he's not really enjoying it. the person engaged in a game. person enjoy at celebration.
**D** Yes, he looks like he has a good hand. the person engaged in a game. the person is watching left hand.
**Answer: D** Yes, he looks like he has a good hand.

Figure 8: An example of `OpenVik` context enrichment on the VCR task (See Appendix G.3 for more).

| Method | Accuracy | Precision | Recall | F$_1$ |
|---|---|---|---|---|
| R2C | 56.66 | 56.73 | 56.72 | 56.72 |
| OpenVik + R2C | 59.96 | 60.01 | 60.03 | 60.02 |
| BLIP | 62.50 | 62.50 | 62.45 | 62.47 |
| OpenVik + BLIP | 67.40 | 67.54 | 67.43 | 67.48 |

Table 7: Visual commonsense reasoning results (%) of `OpenVik` context enrichment compared with zero-shot baselines.

**Qualitative examples.** Figure 8 presents an example before and after applying the two-level visual knowledge-based enrichment for visual commonsense reasoning. The results indicate that visual knowledge enhances the correspondence between the correct answer and the image itself.

**Quantitative results.** We assembled a test set of 939 samples from the validation set of the VCR dataset [62]. Each sample in this test set contains questions and answers with a minimum of five nouns and two relations, guaranteeing an adequate level of information complexity for meaningful engagement with open visual knowledge. The results can be found in Table 7. We observe that the enriched visual knowledge helps especially when solving reasoning questions on humans and their interactions with visually impressive entities, such as "*game*" in Figure 8. This enhancement results in a performance improvement above 3.0% over the zero-shot baseline.

## 6 Conclusion, Limitations, and Future Work

This work is the first exploration of a new paradigm of open visual knowledge extraction, which combines an open relational region detector to flexibly pinpoint relational regions and a format-free visual knowledge generator that generates visual knowledge by prompting a multimodality model conditioned on the region of interest. To further enhance the diversity of the generated knowledge, we explore two distinct data enhancement techniques. Extensive knowledge evaluations underscore the correctness and uniqueness of our extracted open visual knowledge, and the consistent improvements observed across various visual reasoning tasks highlight the real-world applicability of `OpenVik`.

While our approach has been shown effective in various scenarios, its performance at larger scales or on more diverse datasets remains to be studied. Future work could investigate its effectiveness across a broader range of tasks and contexts. Also, the current model requires fine-tuning for the visual knowledge extractor. Developing a model that can generalize well with prompt tuning or demonstration augmentation could be another interesting direction for future work.

## 7 Acknowledgments

Carl Yang was supported by the National Institute Of Diabetes And Digestive And Kidney Diseases of the National Institutes of Health under Award Number K25DK135913.

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

# A    Details of Data Augmentation with External Knowledge Resources

✔ *Enhance Relation Recognition*: We enriched the relationships between objects parsed from the original knowledge descriptions by leveraging the external resource of ConceptNet. ConceptNet comprises commonly observed entities and their connections, where edge weights signify the reliability and frequency of these relationships. The typical value of edge weights in ConceptNet is 1. To prevent the redundancy of common information and to maintain the validity of the enriched relations, we categorized the relationships based on their weights. Relationships with weights less than 1 were deemed "weak" and those with a weight of 1 were labeled "average". We refrained from using these categories for relation enhancement. Instead, only relationships with weights greater than 1, indicative of high reliability, were employed for augmenting the relations.

✔ *Boost Entity Perception*: On the entity side, we augment complement entities and descriptive information with two external knowledge resources. On one hand, for descriptions with a high TF-IDF+ score, we enrich related entities of the object from ConceptNet to create additional knowledge descriptions. The relatedness is based on the between-word relatedness score provided by ConceptNet and we take the threshold as 0.85. On the other hand, we employ the Commonsense Transformers (COMET) [4] model to enrich related new objects and descriptive information. The COMET model is a language model designed to generate commonsense knowledge and understand causal relationships between descriptions. It is pretrained using the atomic dataset, which consists of structured, crowd-sourced knowledge about everyday events and their associated causes and effects. The COMET model can provide neighbor descriptions of the given input of nine different categories of relation. We take the `xAttr` and `oEffect` relation categories and augmented the COMET model by formulating the existing knowledge description texts as the input and choose the corresponding category branch during generation for enriching objects and descriptions respectively.

# B    Dataset Information

Table 8: Dataset statistics.

| split | #image | #descriptor | #relation | #subject & object |
|---|---|---|---|---|
| Train | 75,456 | 832,351 | 30,241 | 302,735 |
| Validation | 4,871 | 64,137 | 5,164 | 34,177 |
| Test | 4,873 | 62,579 | 5,031 | 32,384 |

The statistic information of our augmented dataset is summarized in Table 8, where **split** specifies the dataset split, **#image** indicates the number of images in the split, **#descriptor** indicates the total number of relational descriptors of the images, **#relation** is the total number of unique relations in the relational descriptors after deduplication, and **#subject & object** is the total number of subjects and objects contained in the description text.

# C    Implementation Details

| Hyperparameter | Assignment |
|---|---|
| batch size | 4 |
| learning rate optimizer | Adam |
| Adam epsilon | 1e-8 |
| Adam initial learning rate | 1e-5 |
| learning rate scheduler | cosine scheduler |
| Adam decay weight | 0.05 |

Table 9: Hyperparameters for training open relational region detector.

| Hyperparameter | Assignment |
|---|---|
| batch size | 4 |
| learning rate optimizer | Adam |
| Adam epsilon | 1e-8 |
| Adam initial learning rate | 1e-5 |
| learning rate scheduler | cosine scheduler |
| Adam decay weight | 0.05 |
| $\alpha$ | 0.7 |
| $\phi$ | 0.01 |

Table 10: Hyperparameters for training format-free visual knowledge generator.

**Open relational region detector.** The visual feature extraction backbone is constructed upon a pre-trained ResNet50-FPN. The detector head incorporates a $BLIP_{base}$ equipped with the essential

ViT-B/16 for text supervision, using multiple fully connected layers to derive region features. For each candidate region, we engage a regressor to conduct boundary regression on these features. The detector undergoes fine-tuning for 20 epochs using the relational region bounding box dataset and an Adam optimizer [30]. The hyperparameters for training are detailed in Table 9.

**Format-free visual knowledge generator.** The format-free visual knowledge generator is initialized from BLIP$_{base}$, which incorporates the basic ViT-B/16. We fine-tune the generator model for 20 epochs using the same optimizer as the one employed for the region detector. Detailed hyperparameters for the visual knowledge generator can be found in Table 10.

## D Human Evaluation Guidance and Interface

We perform the human evaluation on two of the four in-depth knowledge quality assessment metrics. We build an interface by referring to [50], where raters are presented with a given image and the corresponding knowledge descriptions and are required to choose one from the multiple choice for two questions on whether the knowledge is valid to humans and whether the knowledge description depicts the image. The detailed scoring criteria for *Validity* and *Conformity* are provided below:

- *Validity* (↑): *whether the generated visual knowledge is valid to humans*.
    - 0 (Invalid): The knowledge description does not conform to human cognition, rendering it unreliable or misleading to humans.
    - 1 (Valid): The knowledge description is valid and accurately conforms to human cognition, providing reliable and meaningful knowledge to humans.
- *Conformity* (↑): *whether the generated knowledge faithfully depicts the scenarios in the images*.
    - 0 (Inconsistent): The knowledge description does not faithfully depict the scenarios in the images, showing significant deviations or discrepancies, making it difficult for users to relate the textual information to the visual context.
    - 1 (Partially Conforming): The knowledge description partially conforms to the scenarios in the images, but there might be minor inconsistencies or missing relevant details.
    - 2 (Moderately Conforming): The knowledge description exhibits a moderate level of conformity with the scenarios in the images, capturing the key aspects and providing coherent descriptions.
    - 3 (Highly Conforming): The knowledge description highly conforms to the scenarios in the images, accurately capturing the details and faithfully representing the visual context.

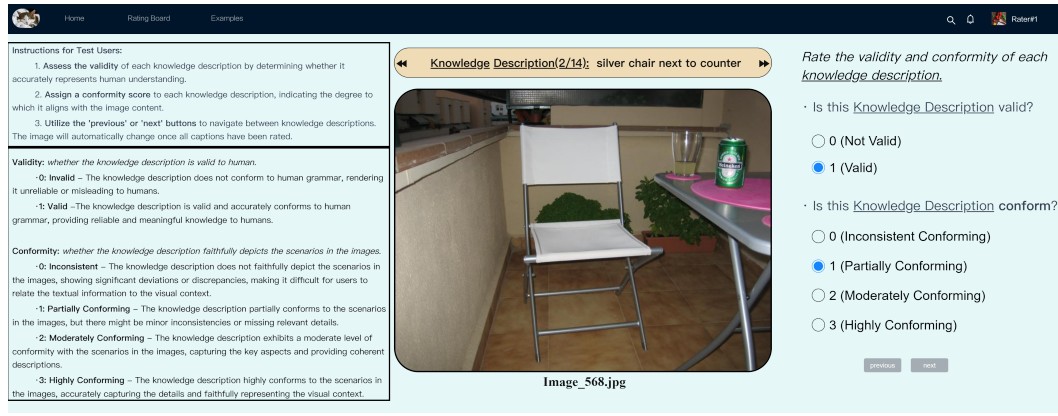

Figure 9: The human evaluation interface for in-depth knowledge quality evaluation.

**Agreement/validation** We use Cohen's $\kappa$ as the agreement score to measure potential subjectivity involved in ratings of knowledge quality. Cohen's $\kappa$ is a statistic that is used to measure inter-rater reliability for qualitative items and is scaled from -1 (perfect systematic disagreement) to 1 (perfect agreement), where values $\leq 0$ as indicating *no agreement* and 0.01-0.20 as *none to slight*, 0.21-0.40 as *fair*, 0.41–0.60 as *moderate*, 0.61-0.80 as *substantial*, and 0.81-1.00 as *almost perfect* agreement. Our calculated average pairwise Cohen's $\kappa$ on human evaluation results from three different raters is 0.76, which indicates a good agreement.

# E  Parametric Knowledge Prompting Template

Given an image $\mathcal{I}$ and the corresponding extracted visual knowledge from it based on `OpenVik`, we perform knowledge comparison with parametric knowledge contained in LLM by prompting the gpt-3.5-turbo model with the object information contained in the $\mathcal{I}$. The prompt format is shown in the followings:

```
Suppose you are looking at an image that contains the following subject
and object entities:
Subject list:  [Insert the subject names here]
Object list:  [Insert the object names here]
Please extract 5-10 condensed descriptions that describe the interactions
and/or relations among those entities in the image.  Try to elucidate the
associations and relationships with diverse language formats instead of
being restricted to sub-verb-obj tuples.
```

# F  More Case Studies of Open Visual Knowledge from `OpenVik`

Figure 10 shows some other cases on the extracted open visual knowledge from `OpenVik`. In comparison to VG and Relational Caps, `OpenVik` exhibits superior performance at capturing novel entities, expanding object interactions through diverse relations, and enriching knowledge representation with nuanced descriptive details. For example for the bottom right image, `OpenVik` can extract novel entities such as " *tracks* ", " *shoe* ", diverse relations such as " *sticking out of* ", and nuanced descriptive details such as " *cold thick* ", " *with man feet on it* ", " *brave* ". The generated knowledge with a more format-free semantic structure is highlighted in red.

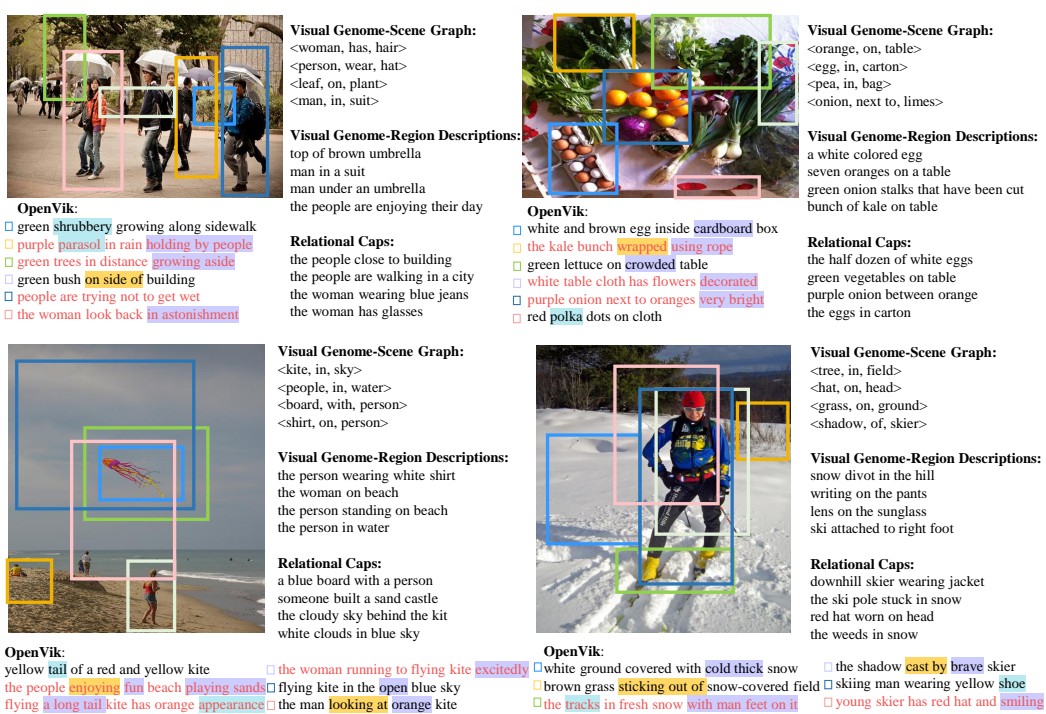

Figure 10: Case studies of open visual knowledge from `OpenVik`.

# G   More Qualitative Examples on Applications

## G.1   Text-to-Image Retrieval

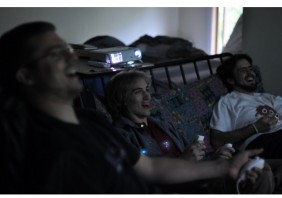

**Original text:** Three young men playing Wii on a projection television. Three men laughing at some pictures from a projector. A group of gentleman playing video games in a dimly lit room. Some people chilling on the couch playing with a Nintendo Wii. A group of men playing a game with remote controllers.

**Enriched text:** men in group. men behind people. men playing. men in room playing video game. group of people. men in group are playing video game. people playing. people watching game. playing game.

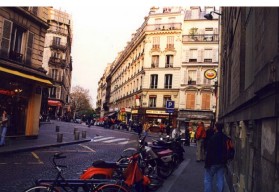

**Original text:** A row of parked motorcycles sitting in front of a tall building. A stone street with bicycles and motor bikes parked on the side and people standing on the sidewalks in front of buildings. Cityscape of pedestrians enjoying an old European city. a row of bikes and mopeds is parked along the street. Motorcycles and mopeds line a side street during the day in a city.

**Enriched text:** row made of stone leading into city. motor in row. row of people. street made of stone. wall made of stone next to side. stone wall behind people. people in line crossing street. street in city. motor on side. people riding motor in city. motor in line. people in line in city. day at city.

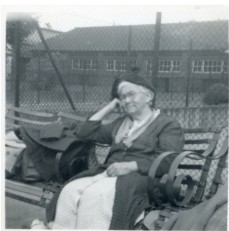

**Original text:** An elderly woman sitting on the bench resting. An old woman leans on her back while sitting on an ornate bench. A woman is sitting on a bench near a fence. Older woman in dress sitting on a park bench. An old woman sitting on a bench next to a fence.

**Enriched text:** woman sitting on bench with a ornate. woman behind fence. woman wearing dress. woman in park. bench by fence. bench in park. woman in ornate dress on the bench. fence behind park.

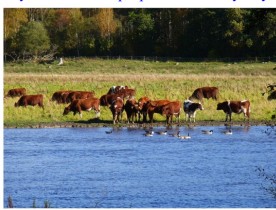

**Original text:** A herd of cattle is feeding at the river's edge. Many cows next to a body of water in a field. A herd of cows grazes in a field near a river. A herd of cattle standing in grassy area next to water. A herd of cattle is near a flock of birds swimming in the water.

**Enriched text:** herd of cattle crossing river. herd traveling by water. cattle crossing river. cattle in field. river across field in front of area. water near field. water near area. water next to flock. Birds inside of water. flock in field.

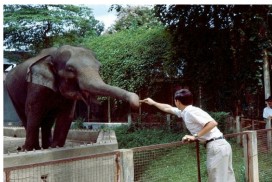

**Original text:** A man is leaning over a fence offering food to an elephant. A man reaching out to an elephants trunk near a gate. A man is feeding an elephant over a fence. A man handing an elephant a stick in an enclosure at a zoo. A man reaches out to give the elephant something.

**Enriched test:** man behind fence. man next to trunk preparing food. man holding stick in enclosure. man pointing at something. fence truck behind food. fence wrapped around trunk. fence behind elephant. fence made of stick. fence surrounds enclosure. trunk of elephant. elephant in enclosure.

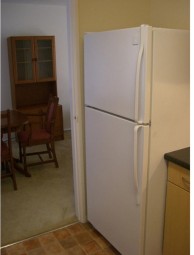

**Original text:** A white refrigerator freezer sitting inside of a kitchen. A corner of a kitchen with a big fridge. A kitchen has a plain white fridge in the corner. A refrigerator in the corner of a kitchen just off the dining room a room showing a very big fridge and a dining table.

**Enriched text:** refrigerator has freezer. refrigerator in corner. refrigerator in bright kitchen. refrigerator in room. refrigerator next to table sitting in kitchen. freezer next to table. corner window in room. corner of table. fridge in kitchen. table in kitchen. fridge table next to table in room.

Figure 11: Qualitative examples of `OpenVik` context enrichment on text-to-image retrieval.

Figure 11 presents more qualitative examples of `OpenVik`-based visual knowledge enrichment on captions. The enriched text is based on the objects present in the images themselves, supplemented with additional relationships from our generated visual knowledge in `OpenVik`. It is shown that the introduced relationships often provide new context information that aligns with the visual content of the images. For example, in the image of an old woman sitting on a bench in a park, the enriched context information includes the positional relationship between the "*bench*", "*fence*", and "*park*", which provides a more comprehensive description of the original image.

## G.2   Grounded Situation Recognition

Figure 12 presents more qualitative examples of `OpenVik`-based context enrichment in the grounded situation recognition (GSR) task. Our context enrichment setting for the GSR task is to perform enrichment based on verbs like "*shopping*" and "*carrying*". We further restrict the enriched context with the objects contained in the image to avoid noisy enrichment. For example, for the image showing people shopping at a market, the enriched knowledge contexts could be "*the people shopping at market*", "*standing person shopping for fruit*". The idea is to enrich the original description $\mathcal{T}$: "*An image of <verb>*" with relevant actions and relations with the extracted visual knowledge from `OpenVik`, which can potentially help in drawing-in the matched candidates.

## G.3   Visual Commonsense Reasoning

Figure 13 presents more qualitative examples of `OpenVik`-based context enrichment in the visual commonsense reasoning (VCR) task. The context enrichment on VCR is performed at two-level,

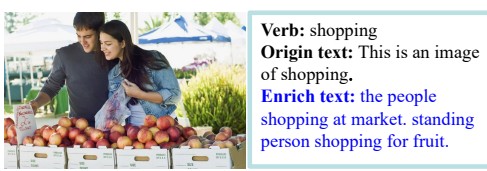 **Verb:** shopping
**Origin text:** This is an image of shopping.
**Enrich text:** the people shopping at market. standing person shopping for fruit.

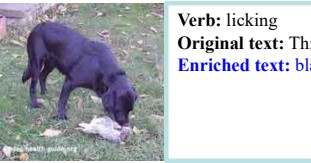 **Verb:** licking
**Original text:** This is an image of licking.
**Enriched text:** black dog licking food.

**Verb:** carving
**Original text:** This is an image of carving.
**Enriched text:** wood carving in center. man carving wood.
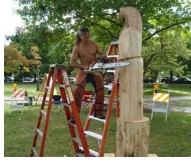

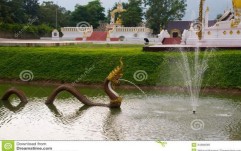 **Verb:** spraying
**Original text:** This is an image of spraying.
**Enriched text:** the water spraying from fountain. the water spraying from spout. the water spraying in park.

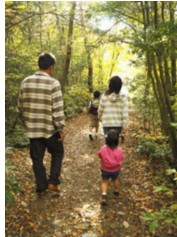 **Verb:** walking
**Original text:** This is an image of walking.
**Enriched text:** the person walking through forest. the people walking on sidewalk.

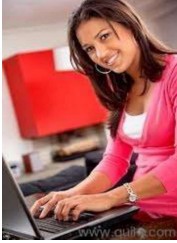 **Verb:** typing
**Original text:** This is an image of typing.
**Enriched text:** sitting woman typing on smart open laptop.

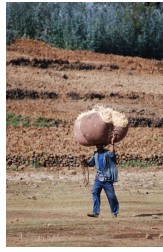 **Verb:** carrying
**Original text:** This is an image of carrying.
**Enriched text:** walking person carrying bag. man carrying hay in the field.

Figure 12: Qualitative examples of `OpenVik` context enrichment on task GSR.

incorporating both entities and relations: (1) we parse the question and options to obtain all (S, O) pairs and, for each entity pair, apply the same relation augmentation as in the image retrieval task; (2) for the V in each option, we enrich the visual context using the same method as illustrated in GSR. It is shown that unrelated answers are usually enriched with contexts that are not relevant to the image, thus enlarging the distance between incorrect answers and the question, e.g., the enriched contexts "*squating person fixing handy bathroom*" for example 3 in Figure 13. At the same time, the knowledge description of the correct answer is enhanced by incorporating information that aligns with the image contents, e.g., the enriched knowledge contexts "*sitting people on red ground*" for example 1 in Figure 13.

## H  Full List of Filtered Verbs for GSR

We provide the full list of verbs out of the predefined 504 candidates of GSR [36] that can be accurate-matched or fuzzy-matched to extracted visual knowledge in Table 11, based on which we compose the testing subset for our evaluation on GSR application in Section 5.2.

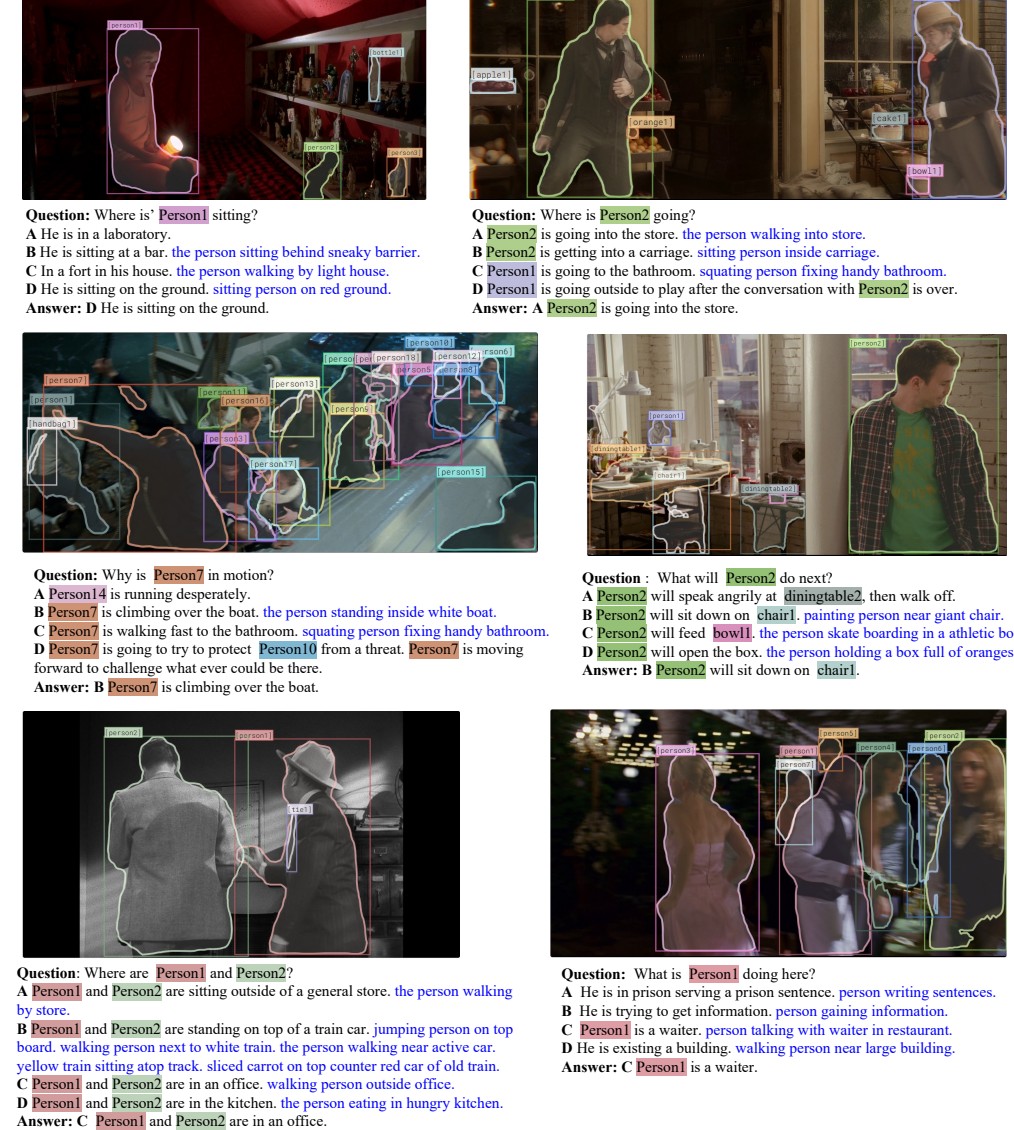

**Question:** Where is' Person1 sitting?
A He is in a laboratory.
B He is sitting at a bar. the person sitting behind sneaky barrier.
C In a fort in his house. the person walking by light house.
D He is sitting on the ground. sitting person on red ground.
**Answer: D** He is sitting on the ground.

**Question:** Where is Person2 going?
A Person2 is going into the store. the person walking into store.
B Person2 is getting into a carriage. sitting person inside carriage.
C Person1 is going to the bathroom. squating person fixing handy bathroom.
D Person1 is going outside to play after the conversation with Person2 is over.
**Answer: A** Person2 is going into the store.

**Question:** Why is Person7 in motion?
A Person14 is running desperately.
B Person7 is climbing over the boat. the person standing inside white boat.
C Person7 is walking fast to the bathroom. squating person fixing handy bathroom.
D Person7 is going to try to protect Person10 from a threat. Person7 is moving forward to challenge what ever could be there.
**Answer: B** Person7 is climbing over the boat.

**Question :** What will Person2 do next?
A Person2 will speak angrily at diningtable2, then walk off.
B Person2 will sit down on chair1. painting person near giant chair.
C Person2 will feed bowl1. the person skate boarding in a athletic bowl.
D Person2 will open the box. the person holding a box full of oranges.
**Answer: B** Person2 will sit down on chair1.

**Question:** Where are Person1 and Person2?
A Person1 and Person2 are sitting outside of a general store. the person walking by store.
B Person1 and Person2 are standing on top of a train car. jumping person on top board. walking person next to white train. the person walking near active car. yellow train sitting atop track. sliced carrot on top counter red car of old train.
C Person1 and Person2 are in an office. walking person outside office.
D Person1 and Person2 are in the kitchen. the person eating in hungry kitchen.
**Answer: C** Person1 and Person2 are in an office.

**Question:** What is Person1 doing here?
A He is in prison serving a prison sentence. person writing sentences.
B He is trying to get information. person gaining information.
C Person1 is a waiter. person talking with waiter in restaurant.
D He is existing a building. walking person near large building.
**Answer: C** Person1 is a waiter.

Figure 13: Qualitative examples of OpenVik context enrichment on task VCR.

Table 11: The full list of filtered verbs for GSR.

| Matching Type | The Word List of Event Types |
|---|---|
| *Accurate* | putting, butting, bathing, dusting, rearing, turning, skating, placing, carting, staring, biting, mashing, folding, wetting, sprinkling, branching, drying, standing, flaming, taxiing, performing, circling, molding, parachuting, glowing, fishing, drinking, speaking, pawing, blocking, milking, racing, stripping, potting, spinning, eating, making, kicking, catching, lacing, urinating, sleeping, pressing, buttering, shearing, sliding, hiking, glaring, dipping, swimming, shopping, slicing, shelling, wagging, grilling, crafting, raining, clawing, splashing, rubbing, snowing, breaking, guarding, clipping, sewing, braiding, telephoning, buttoning, waiting, serving, picking, camping, leaning, working, kissing, wrapping, trimming, tripping, pasting, soaring, driving, kneeling, pumping, coloring, lighting, training, ducking, bowing, arching, cooking, checking, pushing, flipping, rocking, cresting, cleaning, reading, nailing, stitching, building, climbing, covering, shelving, attaching, calming, selling, gluing, dyeing, lapping, photographing, peeling, sprouting, licking, displaying, combing, stacking, planting, fastening, buying, mopping, burning, erasing, measuring, dining, tattooing, gardening, decorating, clearing, fixing, weeding, pulling, feeding, watering, crowning, shaking, dripping, emptying, typing, chasing, poking, leaping, pouring, hanging, sniffing, piloting, falling, overflowing, resting, crashing, carving, ballooning, wading, loading, shaving, boarding, pinning, rowing, juggling, shoveling, hugging, throwing, calling, singing, carrying, walking, writing, crouching, floating, painting, opening, tying, riding, strapping, dialing, saying, bubbling, signing, camouflaging, operating, leading, laughing, parading, skiing, drawing, gnawing, celebrating, spreading, filling, giving, running, smelling, plowing, helping, brushing, scooping, adjusting, wrinkling, steering, biking, smiling, spraying, boating, paying, chewing, stuffing, clinging, landing, wheeling, talking, scoring, teaching, jogging, pitching, flapping, tipping, scrubbing, sitting, surfing, stirring, competing, drumming, jumping, filming, dancing, waxing, hitting, recording, baking, waving, washing, signaling, chopping, stretching, rafting, microwaving, phoning, lifting, swinging, releasing, ramming, towing, packing, hauling, frying (*244 words*) |
| *Fuzzy* | educating, marching, spanking, descending, smearing, heaving, cramming, inflating, stooping, inserting, squeezing, tugging, tilting, moistening, swarming, subduing, waddling, winking, flexing, punching, attacking, nuzzling, sprinting, sucking, puckering, sketching, rotting, videotaping, complaining, tuning, locking, hurling, pricking, arranging, constructing, slapping, sweeping, restraining, dousing, frisking, twisting, wringing, hoisting, immersing, shredding, blossoming, igniting, spying, offering, pouting, confronting, docking, assembling, prying, grinning, sharpening, pruning, disciplining, nipping, coaching, nagging, storming, handcuffing, apprehending, bouncing, clenching, taping, distributing, striking, studying, plunging, curling, aiming, sowing, grinding, rinsing, punting, mowing, hitchhiking, skipping, leaking, providing, hunching, spoiling, kneading, burying, foraging, lathering, vaulting, ejecting, mending, pinching, deflecting, ascending, peeing, bothering, repairing, pedaling, ailing, fueling, skidding, scraping, soaking, grimacing, scolding, spitting, knocking, crushing, bandaging, saluting, fording, stumbling, discussing, raking, launching, whirling, fetching, brawling, retrieving, snuggling, exercising, colliding, stroking, whipping, tilling, betting, farming, browsing, examining, dropping, barbecuing, ignoring, asking, flinging, perspiring, embracing, slipping, flicking, smashing, arresting, lecturing, tearing, gasping, applying, counting, spilling, dragging, recovering, practicing, scratching, shooting, packaging, hunting, stinging (*154 words*) |

