# OpenReview forum: "Open Visual Knowledge Extraction via Relation-Oriented Multimodality Model Prompting"
_NeurIPS.cc/2023/Conference — NeurIPS 2023 poster_

### Official Review · Reviewer_QgT9 · 2023-07-05

**Soundness:** 3 good
**Presentation:** 3 good
**Contribution:** 3 good
**Rating:** 7
**Confidence:** 4

**Summary:**

The authors claim that they propose OpenVik which consists of an open relational region detector to detect regions potentially containing relational knowledge and a visual knowledge generator that generates format-free knowledge by prompting the large multimodality model with the detected region of interest. However, some issues are concerned.

**Strengths:**

The authors claim that they propose OpenVik which consists of an open relational region detector to detect regions potentially containing relational knowledge and a visual knowledge generator that generates format-free knowledge by prompting the large multimodality model with the detected region of interest.

**Weaknesses:**

1. Paper has weak technical depth, it requires more technical details.

2. Further experimental analysis needed. Please give qualitative validation.

**Questions:**

1. Paper has weak technical depth, it requires more technical details.

2. Further experimental analysis needed. Please give qualitative validation.

**Limitations:**

Yes.

---

> ### Author Rebuttal · Authors · 2023-08-09
>
> ***Technical depth and details***
>
> **A**: We would like to stress several distinct technical contributions of OpenVik compared with existing models:
>
> - **Open relational region detector**: Existing detectors often focus on locating objects, while OpenVik is trained to directly detect regions that capture **interactions** and **abstract semantic structures**, such as vivid verbs like “attached to” and nuanced details like “full of”. Some alternative region detectors often need additional visual controls, such as mouse clicks [1, 2] or language controls [3] on a combination of predefined sets of textual properties. OpenVik saves this additional input with its automatic visual grounding ability, which thus improves **knowledge diversity and freshness** significantly.
>
> - **Knowledge generator**: One big difference is that OpenVik is the reasoning-driven generation. Existing regional captioners or knowledge generators often rely on object-level annotation, where the text decoder generates descriptions based on the localized object set [4]. **This leads to the model working like a bag-of-word and a lack of deep semantic understanding.** OpenVik provides **better knowledge grounding** by conditioning the generator on the detected relational region. It includes the ability to **automatically discover the types of interest** that are not only salient but also benefit downstream relational reasoning, as proved in various downstream tasks in Section 5.
>
> - **Language diversity**: Training such a new paradigm for open relational visual knowledge extraction is not trivial. The lack of diverse training data and distribution bias present significant challenges. As shown in Section 4.3 and Figure 3, the **diversity-driven data enhancement** strategies put forth in our work can effectively optimize knowledge richness. They are also generalizable to other models or backbones.
>
> In summary, our work has attempted to explore uncharted territory in the field, with extensive knowledge evaluation and a variety of downstream tasks demonstrating its effectiveness.
>
> ***Further experimental analysis and qualitative validation***
>
> **A**: Thank you for emphasizing the importance of experimental analysis, particularly with respect to qualitative validation. In our study, we have conducted a comprehensive evaluation of the proposed open visual knowledge extraction pipeline. This includes internal assessment of knowledge quality and external evaluation of downstream tasks.
>
> - **Internal Knowledge Quality Evaluation**: We assessed the knowledge generation performance using traditional generative metrics and conducted an in-depth analysis of knowledge quality. Moreover, we compared our approach with existing knowledge sources, as illustrated in Figure 2.
>
> - **External Downstream Tasks Evaluation**: To gauge whether OpenVik's extracted open visual knowledge can enhance reasoning and inference capabilities in various applications, we explored three multimodal tasks, detailed in Section 5. Our analysis included extensive qualitative and quantitative assessments of the results.
>
> - **Qualitative Validation**: We also recognized the importance of providing concrete examples to illustrate our findings. Therefore, Appendix F included a case study featuring qualitative examples of Open Visual Knowledge from OpenVik. More qualitative illustrations from the three downstream applications are available in Appendix G.
>
> By combining these different methods of evaluation, our study provides a well-rounded perspective on the effectiveness and potential applications of OpenVik. We appreciate the valuable feedback and will ensure that the final version of our paper highlights these aspects in a clear and compelling manner.

---

> > ### Author Response · Authors · 2023-08-21
> >
> > Dear reviewer QgT9,
> >
> > We hope our detailed rebuttal has addressed some of your concerns. As we are getting really close to the deadline of the discussion phase, we would really appreciate it if you could kindly give us a chance to further address your questions and/or concerns if any.
> >
> >
> > Many thanks,
> > Authors

---

> > ### Comment · Reviewer_QgT9 · 2023-08-21
> >
> > Thanks for the detailed explanation to my questions. I've carefully read your reply and other reviewers' comments. Therefore, I've changed my rating from 6 to 7.

---

> > > ### Author Response · Authors · 2023-08-21
> > >
> > > Dear reviewer QgT9,
> > >
> > > Thank you for raising the score and we will make sure to properly incorporate the additional discussions in the rebuttal into our revision.

---

### Official Review · Reviewer_nGQ3 · 2023-07-06

**Soundness:** 3 good
**Presentation:** 3 good
**Contribution:** 2 fair
**Rating:** 4
**Confidence:** 4

**Summary:**

Authors present OpenVik, a method for open visual knowledge extraction. It consists of an open relational region detector to detect regions potentially containing relational knowledge and a visual knowledge generator that generates format-free knowledge by prompting the large multimodality model with the detected region of interest. They explore two data enhancement techniques for diversifying the generated format-free visual knowledge. Extensive knowledge quality evaluations highlight the correctness and uniqueness of the extracted open visual knowledge by OpenVik. Moreover, they show consistent improvements across various visual reasoning applications, indicating the real-world applicability of OpenVik.

**Strengths:**

- Well-written
- Extensive Evaluations (Ablation Study and Applications)
- OpenVik outperforms traditional approaches and generates rich Scene Graphs


**Weaknesses:**

- The importance of the pre-training for the Open Relational Region Detector is mentioned but not evaluated

- It seems that the performance of OpenVik is mainly based on BLIP's performance
Ablation of OpenVik vs. BLIP is missing.

- Usefulness of Scene Graph needs to be evaluated instead of diversity

**Questions:**

- How useful are the generated facts?
There seems to be a large variance in the entities, relations and the overall details in the facts. When looking at the produced facts, it can be seen that the facts generated for individual image patches could be quite useless and random, e.g. "dark brown mane growing behind head", and also not related to the main entities in the image, e.g. zebra.
There is no real structure in the facts.

- Do the generated facts suffer from typical problems like hallucination?
If this is the case authors should also elaborate on that.


**Limitations:**

The paper should elaborate more on the difficulties and the limitations of their approach.

---

> ### Author Rebuttal · Authors · 2023-08-09
>
> >***The importance of the pre-training for the Open Relational Region Detector is mentioned but not evaluated***
>
> **A**: Thanks for the observation. Please refer to our added results under ***Q2*** in the global response.
>
> >***Ablation of OpenVik vs. BLIP***
>
> **A**: While BLIP primarily focuses on whole image captioning, making it not directly comparable to OpenVik's relational knowledge generation. We take the relational regions detected by our trained region detector as the input for BLIP decoder. Please refer to our updated results under ***Q1*** in the global response.
>
> >***Usefulness of Scene Graph needs to be evaluated instead of diversity***
>
> **A**: Besides the internal knowledge quality evaluation, such as diversity, three downstream external tasks in Section 5 extend beyond these measures.
>
> These tasks are designed to provide external validation of the knoowledge effectiveness generated by OpenVik and its capability to enhance downstream tasks. We follow a zero-shot setting, while this augmentation can also be conducted with other backbones. Results demonstrate that the backbone model significantly benefits from the context provided by OpenVik, underscoring the value of the extracted visual knowledge.
>
> To demonstrate generalizability, we expanded our comparisons to include an additional contemporary backbone variant (BLIP, R2C [1]) for each application, as shown below (See full Tables 13-15 in the attached PDF). The results confirm that this added model benefits from OpenVik's contexts, similar to the zero-shot ones.
>
> |  Method  | Recall@1  | Recall@5 | Recall@10 | Avg |
> | -------------------- | ----- | ------- | ------ | -------- |
> | BLIP | 63.11 | 86.30  | 91.10  | 80.17    |
> | OpenVik + BLIP  | 65.23 | 87.71 | 91.90  | 81.61  |
>
> |  Method  | Accuracy  | Precision | Recall | F1 |
> | -------------------- | ----- | ------- | ------ | -------- |
> | BLIP | 70.42 | 65.32 | 69.25 | 67.23  |
> | OpenVik + BLIP  | 80.25 | 72.55 | 70.61  | 71.57  |
>
>  Method       | Accuracy | Precision | Recall | F1    |
> | ----------------- | -------- | --------- | ------ | ----- |
> | R2C            | 62.50    | 62.50     | 62.45  | 62.47 |
> | OpenVik + R2C     | 67.40    | 67.54     | 67.43  | 67.48 |
>
> [1] Zellers, Rowan, et al. "From recognition to cognition: Visual commonsense reasoning." CVPR, 2019.
>
> >***How useful are the generated facts? the facts could be quite useless and random, e.g. "dark brown mane growing behind head", and also not related to the main entities in the image, e.g. zebra. There is no real structure in the facts.***
>
> **A**: Thank you for your insightful observation regarding the variance in the generated facts. It is essential to understand that the task output (such as "dark brown mane growing behind head") is intentionally not following the conventional SVO structure. Our goal is to explore and capture nuanced visual semantics in a more flexible and unconstrained manner, without being restricted by fixed formats. Let's break down the semantics of this example:
>
> - Subject: "mane" (referring to the hair on the neck of a mammal like a horse, lion, or zebra)
> - Adjectives: "dark brown" (specifying the color of the mane)
> - Verb: "growing" (describing the appearance or position of the mane)
> - Prepositional Phrase: "behind head" (providing the location of the mane)
>
> This description offers a detailed and relevant depiction of the main entity, focusing on attributes such as color and position, and does not adhere strictly to the SVO structure.
>
> Visual relational knowledge, as opposed to traditional text-based extraction, emphasizes intricate details like tools, sizes, and spatial relationships. By capturing these complexities, our approach complements existing knowledge literature established on texts, enhancing logical reasoning and promoting explainable AI in visual tasks. This can also alleviate hallucination problems in QA/captioning and add a layer of relational regularization to factual knowledge, mitigating bias in large language model prompting.
>
> We do recognize that the language generation process may lead to variations in the quality and structure of the facts generated. This is reflective of the diversity and intricacy found within visual information and indeed points to areas for further refinement in our model. However, this does not diminish the significance of open visual knowledge extraction, which serves as an indispensable component in capturing the visual aspect of world information.
>
> >***Do the generated facts suffer from hallucination***
>
> **A**: Thank you for raising the important question regarding potential hallucinations in knowledge generation. It is insightful to recognize that unbalanced and noisy distributions within the training data can indeed lead to errors in the knowledge produced. If we view hallucinations as unwarranted inferences based on input, then such inaccuracies in OpenVik and comparable baselines are more typically a result of detection errors due to data bias by associating wrong features to a class/label. Two illustrative failure cases can be found in the attached PDF, Figure 1. For example, in the left figure, a ladder has been misidentified as a towel, leading to the erroneous description of a 'blue towel hanging from dry shower.' In the right figure, a 'black speaker by flat tv' is generated, although the speaker is not present in the image—possibly reflecting common co-occurrences within the data set. To solve this problem, the key is identifying the cofounder feature of class labeling.
>
> The unwarranted inference error is more prevalent across QA and captioning tasks, rather than the fact generation process. However, OpenVik is helpful to mitigate the hallucinations in QA and captioning since it can discover unique relational facts. OpenVik's performance metrics demonstrate that it maintains a desirable balance between diversity and validity, offering more accurate grounding that can enhance high-level tasks such as storytelling.

---

> > ### Author Response · Authors · 2023-08-21
> >
> > Dear reviewer nGQ3,
> >
> > We hope our detailed rebuttal has addressed some of your concerns. As we are getting really close to the deadline of the discussion phase, we would really appreciate it if you could kindly give us a chance to further address your questions and/or concerns if any.
> >
> >
> > Many thanks,
> > Authors

---

### Official Review · Reviewer_GNm1 · 2023-07-06

**Soundness:** 2 fair
**Presentation:** 4 excellent
**Contribution:** 2 fair
**Rating:** 7
**Confidence:** 4

**Summary:**

This paper introduces a novel approach to visual knowledge extraction named OpenVik. It comprises three main components: the Open Relational Region Detector, the Format-Free Visual Knowledge Generator, and the Diversity-Driven Data Enhancement module.

The Open Relational Region Detector, built on the object detection framework FasterRCNN, is fine-tuned to discern relation-oriented visual regions in images. The Format-Free Visual Knowledge Generator's purpose is to utilize this relational region information to create more relation-relevant visual knowledge from an image. The last component, the Diversity-Driven Data Enhancement module, functions as a knowledge post-processing module. Its primary goals include filtering out extraneous knowledge facts and enriching relations via synonym substitution, leveraging external knowledge resources.

The experimental results indicate that OpenVik surpasses previous benchmarks in most test scenarios, implying its effectiveness in generating high-quality relational knowledge from images.

**Strengths:**

1. The visual representation of the data through figures and charts is commendable. In particular, Figure 1 effectively illustrates the primary components of OpenVik. The color-coded highlights for 'entities', 'relations', and 'descriptive details' significantly enhance the readability of the examples.

2. The architectural design of the model, along with the selected loss functions, is robust and well-conceived. The three components harmoniously co-function to train on current relational datasets and subsequently generate novel open visual knowledge from images.

3. The level of detail in the explanation of the proposed method is noteworthy. The comprehensive presentation allows readers to gain a thorough understanding of OpenVik even after a single read-through.

**Weaknesses:**

1. The paper could benefit from a stronger justification for its approach. With the existence of established multimodal models like BLIP[1], mPLUG[2], OFA[3], and recent advancements like BLIP-2[4], MiniGPT4[5], all of which can potentially perform Visual Knowledge Extraction through prompting, the specific need for OpenVik is not explicitly clarified. Further explanation about what unique advantages OpenVik brings would be beneficial.

2. The comparative analysis in section 4.1 appears unbalanced. The use of older models, such as the captioning model and relational captioning model, for benchmarking does not provide a robust basis for comparison. It would be more persuasive if contemporary models were included in the comparison.

3. The ablation study appears to be not comprehensive enough. Crucial distinctions, such as those between the Open Relational Region Detector and the standard FasterRCNN, are not sufficiently elucidated. More detailed examinations of such comparisons would significantly enhance the paper's depth of analysis.

4. The paper lacks clear explanation regarding the evaluation settings, such as the specific split used during evaluation. While I did find these details eventually in the supplementary material, I would recommend the authors incorporate such essential information directly within the main body of the paper. This would ensure a smoother reading experience and easier access to important technical specifications.

[1] BLIP: Bootstrapping Language-Image Pre-training for Unified Vision-Language Understanding and Generation

[2] mPLUG: Effective and Efficient Vision-Language Learning by Cross-modal Skip-connections

[3] OFA: Unifying Architectures, Tasks, and Modalities Through a Simple Sequence-to-Sequence Learning Framework

[4] BLIP-2: Bootstrapping Language-Image Pre-training with Frozen Image Encoders and Large Language Models

[5] MiniGPT-4: Enhancing Vision-Language Understanding with Advanced Large Language Models

**Questions:**

1. With the existence of general multimodal models, why do we need OpenVik?

Please answer my questions mentioned in the **Weaknesses** .

**Limitations:**

Good

---

> ### Author Rebuttal · Authors · 2023-08-09
>
> ***What unique advantages OpenVik bring compared with existing multimodal models? With the existence of general multimodal models, why do we need OpenVik***
>
> **A**: Thank you for raising the question regarding the need for OpenVik in the context of existing multimodal models. The specific need for OpenVik lies in addressing key challenges in visual relational knowledge that current models still grapple with:
>
> - **Current Models Exhibit Deficiencies in Compositional Understanding**: Despite the success of large vision and language models (VLMs) in many downstream applications, it is unclear how well they encode compositional information. Recent studies have demonstrated that state-of-the-art VLMs have poor relational understanding and often rely on object-centric shortcuts, leading to the phenomenon that the models behave like a bag of words. This can blunder when linking objects to their attributes, and demonstrate a severe lack of semantic role sensitivity [1, 2]. OpenVik's focus on relational knowledge extraction encourages deep semantic structure understanding, and visual prompting enables semantic grounding across modalities.
>
> - **Relational Knowledge Helps in Complex Planning and Reasoning**: Tasks requiring intricate comprehension, such as planning and reasoning, benefit from relational knowledge [3, 4, 5, 6]. The relational knowledge from OpenVik can enhance compositional generalization in visual reasoning, thereby contributing to the system's trustworthiness and robustness.
>
> - **Openness and Diversity**: OpenVik's specialized approach to open relational detection, format-free knowledge generation, and diversity-driven data enhancement provides unique advantages for discovering nuanced, detailed, verbs that are less seen in the training set.
>
> [1] Yuksekgonul, Mert, et al. "When and why vision-language models behave like bag-of-words models, and what to do about it?." ICLR, 2023.\
> [2] Hendricks, Lisa Anne, and Aida Nematzadeh. "Probing image-language transformers for verb understanding." ACL, 2021.\
> [3] Li, Bo, et al. "Evaluating ChatGPT's Information Extraction Capabilities: An Assessment of Performance, Explainability, Calibration, and Faithfulness." arXiv preprint arXiv:2304.11633 (2023).\
> [4] Wang, Renhao, et al. "Programmatically Grounded, Compositionally Generalizable Robotic Manipulation." ICLR, 2023.\
> [5] Kurenkov, Andrey, et al. "Modeling Dynamic Environments with Scene Graph Memory." International Conference on Machine Learning. PMLR, 2023.\
> [6] Hsu, Joy, Jiayuan Mao, and Jiajun Wu. "DisCo: Improving Compositional Generalization in Visual Reasoning through Distribution Coverage." TMLR, 2022.
>
> ***It would be more persuasive if contemporary models were included in the comparison***
>
> **A**: We thank the reviewer for the insightful comments. Please refer to our added results under ***Q1*** in the global response.
>
> ***Crucial distinctions, such as those between the Open Relational Region Detector and the standard FasterRCNN, are not sufficiently elucidated***
>
> **A**: Please note that a direct comparison between the two may not be straightforward due to their differing objectives. The standard FasterRCNN is designed for object-level detection and classification, wherein the objects are restricted to a predefined set. In contrast, our Open Relational Region Detector is trained to automatically detect relational regions potentially containing relation-oriented knowledge, which is a more complex and nuanced task.
>
> In response to the comments, we have added an ablation study to illustrate the effects of loading a pre-trained detector backbone versus training the detector from scratch without pre-training. Please refer to our results under ***Q2*** in the global response.
>
> ***Incorporate evaluation settings directly within the main body of the paper***
>
> **A**: Thanks for the suggestions! Due to the page limit and the experiment quantity of knowledge generation as well as three downstream tasks, we only included the main module implementation details in the main body, with the specifications for downstream evaluations in Appendix C and D. We appreciate your suggestions on the significance of such details and will ensure that the essential evaluation specifications are included in the main body of the final version.

---

> > ### Comment · Reviewer_GNm1 · 2023-08-19
> > **Reply to Author's Rebuttal**
> >
> > Thanks for the detailed explanation to my questions. I've carefully read your reply and other reviewers' comments. The ablation study in global response do convince me the effectiveness of the proposed model. Therefore, I've changed my rating from 5 to 7.

---

> > > ### Author Response · Authors · 2023-08-20
> > >
> > > We thank the reviewer for the reply and we are happy to note that we have addressed your concerns in the rebuttal. We will make sure to properly incorporate the additional results and discussions in the rebuttal into our revised paper.

---

### Official Review · Reviewer_hCst · 2023-07-07

**Soundness:** 3 good
**Presentation:** 3 good
**Contribution:** 2 fair
**Rating:** 5
**Confidence:** 4

**Summary:**

This paper that introduces a new paradigm of open visual knowledge extraction called OpenVik. This method generates format-free knowledge by prompting a large multimodality model with detected regions of interest. The proposed framework consists of an open relational region detector and a format-free visual knowledge generator. The paper highlights the limitations of existing approaches to visual knowledge extraction and demonstrates the correctness and uniqueness of the extracted open visual knowledge by OpenVik. The extracted knowledge is integrated across various visual reasoning applications, showing consistent improvements and indicating the real-world applicability of OpenVik.

**Strengths:**

1. Novel Approach: The paper introduces a new paradigm, OpenVik, for open visual knowledge extraction. This system generates format-free knowledge by prompting a large multimodality model with detected regions of interest, which is a novel approach in the field.
2. Comprehensive Evaluation: The paper provides a thorough evaluation of the generated knowledge, using traditional generative metrics and in-depth knowledge quality assessment. This comprehensive evaluation helps validate the effectiveness of the proposed method.
3. Comparison with Existing Knowledge Sources: The paper compares the extracted visual knowledge with non-parametric knowledge in existing knowledge graphs and parametric knowledge from large language models. This comparison provides a clear understanding of the unique value proposition of OpenVik.
4. Real-World Application: The extracted knowledge is integrated across various visual reasoning applications, showing consistent improvements. This indicates the real-world applicability of OpenVik, making it a practical solution for visual knowledge extraction.

**Weaknesses:**

1. Limited Dataset: The training data for the model is built based on Visual Genome and its relation-enhanced version Dense Relational Captioning. The performance of the model on more diverse datasets remains to be studied, which could limit its generalizability.
2. Implementation Complexity: The model involves complex implementation details, including the use of a ResNet50-FPN backbone for the open relational region detector and fine-tuning of the visual knowledge extractor. This complexity could make the model difficult to implement and adapt for other researchers or practitioners.
3. Limited comparisons and lack of ablation studies: The authors only compare to a few baselines, but there are many knowledge-enhanced models which need to be compared in the application section.

Other detailed questions are listed in Questions section.

**Questions:**

1. There are many other notions in Figure 1, such as L_k, can the authors explain all of them or list all the equations?
2. What is binary mask (line 113) in the format-free visual knowledge generator?
3. What is the training loss of the method? It is not clear to me, since there are many notions in Figure 1 but not appearing in the paper.
4. Can this method be scale to larger training dataset, or other backbones?
5. In the evaluation of in-depth knowledge quality, the authors mentioned they randomly selected 100 images. How about the compared method, did they use the same images or same number of images? How many raters or annotators in the studies?
6. In the application section, are the results of OpenVik zero-shot or fine-tuning? The implement details are not clear in the applications, and need to compare with more other methods.

**Limitations:**

yes

---

> ### Author Rebuttal · Authors · 2023-08-09
>
> >***The performance of the model on more diverse datasets remains to be studied. Can this method be scaled to larger training dataset, or other backbones***
>
> **A**: In this work, we chose Visual Genome and its relation-enhanced dataset as the benchmark because they are rich in relational region descriptions, making them the most suitable public data available for training a visual relational knowledge extractor, as far as we know in the existing literature.
>
> It's essential to clarify, however, that our proposed methods are not constrained to these specific datasets or architectures. They have been designed to be model-agnostic and can be flexibly adapted to other backbones or tested on more diverse datasets. The challenge of gathering such datasets that are capable of supporting deep-semantic understanding and cross-modality grounding is complex. We recognize this as an important avenue for future exploration and are eager to expand our research based on the current work.
>
> >***Implementation complexity***
>
> **A**: Our model is composed of two main components: the region detector and the knowledge generator. Both leverages established backbones. We've tailored these designs specifically for open relational tasks, ensuring we avoid adding undue model complexity. To facilitate ease of implementation and to encourage further research in this domain, we intend to release our comprehensive implementation code along with trained model checkpoints.
>
> >***Need to compare with more other methods for the applications and implementation details***
>
> **A**: In our work, three application tasks were utilized to evaluate the efficacy of OpenVik's generated knowledge by augmenting the input with generated knowledge for the challenging zero-shot settings. It's important to recognize that this augmentation can also be adapted to different backbones or in varied settings, while the optimization for incorporating knowledge across diverse applications is beyond the scope of this work.
>
> To further showcase generalizability, we have included additional results (Tables 13–15 in the attached PDF) using a contemporary backbone for each application. The results confirm that this new backbone model benefits from OpenVik's contexts, similar to the zero-shot baseline. This finding further substantiates the usefulness of our approach to extracting visual relational knowledge.
>
> We acknowledge and appreciate your valuable feedback, and we agree that these expanded comparisons and clarifications will strengthen the paper.
>
> For those seeking more specific details on the application implementations, please refer to the following:
> - Retrieval: lines 276-279 (zero-shot, ZS) and 300-307 (knowledge augmented, knowledge+)
> - GSR: lines 296-299 (ZS) and lines 300-307 (knowledge+)
> - VCR: lines 323-325 (ZS) and 326-329 (knowledge+)
>
> >***Notations and training loss***
>
> **A**: Thanks for pointing out the oversight that needs further clarification. We have listed all notations in Tables 10 and 11 in the attached PDF and will ensure to include these crucial details in the final version.
>
> Specifically, for the relational region detector, $L_{RD}$ is the regional regression loss from the detector, and $L_{K}$ is the supervision from the ground truth relational knowledge. The training objective $L_l$ of the relational region detector is formulated as $L_v = L_{RD} + L_{K}$.
>
> - **Region Regression**: This part is guided by our created regional box labels, denoted as $U_j$. More precisely, the foreground of these relation-centric region labels is created by taking the union of the object-level bounding boxes of the entities, such as *boat*, *water*, contained in a ground truth region knowledge description $V_j$. This forms the smooth L1 loss $L_{RD}$ [1] for region regression.
>
> - **Knowledge Supervision**: To assist with the refinement of the bounding box, we replaced the object-centric label classification in traditional object detectors with knowledge supervision. A pre-trained generator is finetuned to create the regional description grounded to the given region. This is supervised by the cross entropy loss  $L_K$ with region description $T_j$ [2].
>
> [1] Ross Girshick. Fast r-cnn. ICCV 2015.\
> [2] Li et al. Blip: Bootstrapping language-image pre-training for unified vision-language understanding and generation. ICML, 2022.
>
> >***What is binary mask in the generator***
>
> **A**: Each binary mask represents a detected region from the open region detector, based on which the model would generate region-specific knowledge.
>
> >***Details for in-depth knowledge quality evaluation***
>
> **A**: All the methods compared were based on the same set of 100 randomly selected images to ensure a fair comparison. As detailed in Appendix D lines 561–562, we utilized three different raters for the study. The calculated average pairwise Cohen's κ value is 0.76, demonstrating good agreement.

---

> > ### Comment · Reviewer_hCst · 2023-08-19
> >
> > Dear Authors,
> >
> > Thanks for your detailed responses. But I think all the notations should be more clear and easier to understand, i.e., what do you mean "The training objective $L_l$ of the relational region detector is formulated as $L_v = L_{RD} + L_K$", then how to calculate $L_l$? Also, I think the improvements using BLIP is sort of minor, so does it matter to choose which backbones?
> >
> > However, I think the responses answer some of my questions, and I am happy to raise my score to borderline accept.

---

> > > ### Author Response · Authors · 2023-08-20
> > >
> > > We thank the reviewer for the constructive feedback. We are pleased to note that we could address some of your concerns in the rebuttal.
> > >
> > > We apologize for any confusion regarding the notation. To clarify for the specific one, there was a typo and the training objective $L_v$ of the relational region detector should be expressed as $L_v=L_{RD}+L_K$. As for the calculation of $L_l$ , please refer to equation 2 in the paper. We will check our notations systematically in the revision and make necessary rearrangements.
> > >
> > > On the topic of backbones, our proposed strategies, notably relational-oriented prompting and diversity-driven data enhancement, are designed with versatility in mind. They can be seamlessly integrated with various vision-language backbones, including ALIGN [1], VL-BART(T5) [5], and SimVLM [6], among others. This adaptability underscores the robustness and broad applicability of our approach.
> > >
> > > Once again, thank you for your time and insights. We will make sure to properly incorporate the additional discussions in the rebuttal into our revised paper.
> > >
> > > [1] Jia, Chao, et al. "Scaling up visual and vision-language representation learning with noisy text supervision." ICML, 2021.
> > > [2] Cho, Jaemin, et al. "Unifying vision-and-language tasks via text generation." ICML, 2021.
> > > [3] Wang, Zirui, et al. "Simvlm: Simple visual language model pretraining with weak supervision." ICLR, 2022.

---

### Official Review · Reviewer_PUUs · 2023-07-07

**Soundness:** 2 fair
**Presentation:** 3 good
**Contribution:** 2 fair
**Rating:** 4
**Confidence:** 4

**Summary:**

This paper proposes a new paradigm called open visual knowledge extraction and designs a framework OpenVik to generate format-free knowledge instead of pre-defined format knowledge. The authors also present two data enhancement technologies to ensure the diversity of knowledge. Moreover, the paradigm could also be integrated with other applications to jointly boost the performance.

**Strengths:**

1. The paper is well-written and easy to understand.
2. The proposed open relational region detector is interesting.
3. The evaluation for in-depth knowledge quality is significant.


**Weaknesses:**

1. The originality of the work is incremental. Indeed, both open relational region detector and format-free visual knowledge generator are minor modification existing models.
2. The experiments presented in this paper are insufficient. The comparison baselines should be more explored because the task of this paper is quite similar to region captioning task, such as [1], [2].
3. The word ‘open’ is confusing in the title. The paper could not tackle the open-world knowledge. It is still based on the close-set of objects.
4. The implementation of open relational region detector lacks the detailed information, such as the update of bounding box. The label set raises another question.

References
[1] Zhong, Yiwu, et al. "Comprehensive image captioning via scene graph decomposition." Computer Vision–ECCV 2020: 16th European Conference, Glasgow, UK, August 23–28, 2020, Proceedings, Part XIV 16. Springer International Publishing, 2020.
[2] Ghosh, Shalini, et al. "Generating natural language explanations for visual question answering using scene graphs and visual attention." arXiv preprint arXiv:1902.05715 (2019).


**Questions:**

1. If the supervision is the generated description, how could it guarantee the correspond label for the region that contains the same objects and relations? Is it a few-shot task?
2. More detail of generator should be provided. What is the input of generator? Is the single region or the whole image?

**Limitations:**

Yes.

---

> ### Author Rebuttal · Authors · 2023-08-09
>
> >***The originality of the work is incremental and made minor modifications to existing models***
>
> **A**: We would like to clarify several distinct differences between OpenVik and existing models:
> - **Open relational region detector**: Existing detectors often focus on locating objects, while OpenVik is trained to directly detect regions that capture **interactions** and **abstract semantic structures**, such as vivid verbs like “attached to” and nuanced details like “full of”. Some alternative region detectors often need additional visual controls, such as mouse clicks [1, 2] or language controls [3] on a combination of predefined sets of textual properties. OpenVik saves this additional input with its automatic visual grounding ability, which thus improves **knowledge diversity and freshness** significantly.
>
> - **Knowledge generator**: One big difference is that OpenVik is the reasoning-driven generation. Existing regional captioners or knowledge generators often rely on object-level annotation, where the text decoder generates descriptions based on the localized object set [4]. **This leads to the model working like a bag-of-word and a lack of deep semantic understanding.** OpenVik provides **better knowledge grounding** by conditioning the generator on the detected relational region. It includes the ability to **automatically discover the types of interest** that are not only salient but also benefit downstream relational reasoning, as proved in various downstream tasks in Section 5.
>
> - **Language diversity**: Training such a new paradigm for open relational visual knowledge extraction is not trivial. The lack of diverse training data and distribution bias present significant challenges. As shown in Section 4.3 and Figure 3, the **diversity-driven data enhancement** strategies put forth in our work can effectively optimize knowledge richness. They are also generalizable to other models or backbones.
>
> In light of these, our work goes beyond minor modifications of existing models. We have sought to tackle the challenging problem of open visual knowledge acquisition and deep semantic grounding. The proposed generic techniques can also be applied to other models and datasets.
>
> [1] Pont-Tuset, Jordi, et al. "Connecting vision and language with localized narratives." ECCV, 2020.\
> [2] Yan, Kun, et al. "Control image captioning spatially and temporally." ACL. 2021.\
> [3] Deng, Chaorui, et al. "Length-controllable image captioning." ECCV, 2020.\
> [4] Wu, Jialian, et al. "Grit: A generative region-to-text transformer for object understanding." arXiv preprint arXiv:2212.00280 (2022).
>
>
> >***The comparison baselines should be more explored since the task is quite similar to region captioning***
>
> **A**: We thank the reviewer for the insightful comments. Please refer to our added results under ***Q1*** in the global response.
>
> >***The paper could not tackle open-world knowledge***
>
> **A**: In OpenVik, both objects and relations go beyond a pre-defined set. This is achieved through the design of two core modules:
>
> - Relational detector: The relational detector performs region-level detection instead of object level detection. This is done within a single, holistic detection box supervised by knowledge beneficial for reasoning, allowing for deep semantic and combinational understanding that is not constrained by specific object categories.
>
> - Generative knowledge process: The subsequent knowledge generation is designed to be generative and format-free, not restricted to mere classification or generation around a closed set of objects. This flexibility allows it to capture information in a more open-ended manner, reflecting the true diversity and complexity of real-world visual information.
>
> Additionally, the diversity strategies proposed help to mitigate distribution bias in training data, encouraging the generation of novel knowledge, including in zero- and few-shot scenarios.
>
> >***The implementation of open relational region detector***
>
> **A**: Thanks for pointing out the areas that need further clarification. The training objective of our relational region detector consists of two components: region regression and knowledge supervision, as detailed in lines 106-110.
>
> - **Region Regression**: This part is guided by our newly generated region labels, denoted as $U_j$. As detailed in lines 108-109, the foreground of these relation-centric region labels is created by taking the union of the object-level bounding boxes of the entities, such as *boat*, *water*, contained in a ground truth region knowledge description $T_j$. This forms the region regression loss $L_{RD}$.
>
> - **Knowledge Supervision**: To assist with the refinement of the bounding box, we have replaced the object-centric label classification found in traditional object detectors with knowledge supervision (line 110). A pre-trained generator is fine-tuned to create the regional description for the given region. This is supervised by the ground truth description $T_j$ with loss term $L_K$.
>
> The combined training objective for the relational region detector is $L_v = L_{RD} + L_K$. We recognize that this notion (in Figure 1) was not clearly explained. We are grateful to the reviewer for pointing it out, and we will make this clear in the final version of the paper.
>
> >***What is the input of generator? Is the single region or the whole image?***
>
> **A**: Thank you for the inquiry. OpenVik can handle multiple detected relational regions obtained from the detectors for a given image. As detailed in lines 118-120, each of these regions is processed individually and serves as an input to the generator. Along with the specific region, the generator also receives the ViT representation of the entire image. The detected region is utilized as a binary attention mask, which helps to filter out the background and concentrate solely on one relational foreground at a time.

---

> > ### Comment · Reviewer_PUUs · 2023-08-20
> >
> > Thanks the authors for the responses.
> >
> > Several concerns are explained. However, I still think the paper lacks novelty. The design of region detector could not reflect remarkable innovation compared with alternative region detectors. Moreover, the region detector restricts the potential of this task. The region annotation is created by the union region of objects defined in the prior datasets, which could not enlarge the detect sets. I suppose it still couldn't solve the open visual knowledge issue. Therefore, I insist on my original rating.

---

> > > ### Author Response · Authors · 2023-08-20
> > > **Clarifying the Role of Region Detector in OpenVik and the Major Novelty of This Work**
> > >
> > > Thank you for taking the time to provide further feedback.
> > >
> > > Firstly about the novelty of region detection, we would like to clarify that the region detector is not claimed as a major novelty in this work. Instead, we adapt an appropriate existing method to provide visual grounding and serve our broader objective of open visual knowledge extraction. The major novelty of this work is on the relational condition framework and subsequent design of generative models for open visual knowledge extraction, which can generate relational knowledge not limited to a specific relation vocabulary, as well as the additional designs to further enhance the openness and richness of the generated knowledge.
> > >
> > > Addressing the second concern about the limitation of region detection, the term "open knowledge" in this work means that the knowledge is not confined to entities or relations from a given dataset, which is achieved through a generative model that can continuously accumulate new relational knowledge from different resources. The generative model is firstly pre-trained on large image captioning datasets. To enhance the diversity and novelty of the output knowledge, we leverage external knowledge sources to supplement relation recognition and boost entity perception. As elaborated in Section 3.3, the vocabulary for both entities and relations is enriched using this external knowledge coupled with a commonsense language model. Through this approach, we tackle the open visual knowledge challenge highlighted by the reviewer. Our empirical evaluations, detailed in lines 251-257 and presented in Table 2, highlight OpenVik's capability in extracting visual knowledge of significantly higher diversity compared to benchmarks like Visual Genome and Relational Caps. Region-oriented openness of relational knowledge has not been a concern in this work, which may or may not be a promising direction, and it can be studied in a future work.
> > >
> > > In conclusion, the reviewer’s concerns are mainly centered on OpenVik's region detector. We believe that while it is a fundamental component of visual knowledge extraction, it is relatively apart from our major novelty in this work and should not overshadow the primary innovations of our innovative reasoning-driven designs in the knowledge generator and the data enhancement mechanisms. Together, they pioneer a fresh approach to knowledge extraction towards both openness and diversity. We sincerely hope our explanation provides a clearer understanding of the significance and objectives of our work.

---

### Author Rebuttal · Authors · 2023-08-09

>***Q1. Adding contemporary regional captioning baselines***

**A**: We appreciate the helpful suggestions on adding region captioning baselines. Note that although the proposed task in our paper has some similarities to region captioning, we would like to highlight the crucial difference in OpenViK: **it is designed to automatically detect regions that can be grounded in diverse and fresh relational knowledge, in contrast with traditional methods where the region detectors primarily focus on object-level areas, or the generators are learned on the bag of words over a fixed class of objects**.

Following the suggestion from reviewers PUUs and GNm1, we expanded our comparison baselines to include more region captioning methods, including Sub-GC [1], BLIP [2], and BLIP2 [3]. The results are shown below (A full comparison with regional captioning methods is included in Table 12 of the attached PDF).
|    Method  | BLEU  | ROUGE-L | METEOR | Validity | Conformity | Freshness | Diversity |
| ---------- | ----- | ------- | ------ | -------- | ---------- | --------- | -------- |
| Sub-GC     | 0.272 | 0.263   | 0.221  | 0.892    | 0.871      | 0.795     | 0.547     |
| BLIP       | 0.264 | 0.266   | 0.252  | 0.886    | 0.855      | 0.760     | 0.531     |
| BLIP2      | 0.275 | 0.285   | 0.257  | 0.892    | 0.871      | 0.766     | 0.535     |
| OpenVik  | 0.280 | 0.283   | 0.250  | 0.907    | 0.883      | 0.809     | 0.619     |

It shows that OpenViK has a similar accuracy performance to the recently proposed SOTA approach but is superior in diversity and freshness metrics, which indicates a deep understanding and richer reasoning ability of OpenViK. The comparison also illustrates that OpenVik's specialized designs, including contrastive decoding and knowledge diversity regularizers, confer clear advantages regarding diversity and freshness.

[1] Zhong, Yiwu, et al. "Comprehensive image captioning via scene graph decomposition." ECCV, 2020. \
[2] Li, Junnan, et al. "Blip: Bootstrapping language-image pre-training for unified vision-language understanding and generation." International Conference on Machine Learning. PMLR, 2022. \
[3] Li, Junnan, et al. "Blip-2: Bootstrapping language-image pre-training with frozen image encoders and large language models." arXiv preprint arXiv:2301.12597 (2023).

>***Q2. Ablation on the pre-training for the open relational region detector***

**A**: To address this concern, we have conducted an additional ablation study, contrasting the outcomes when loading a pre-trained detector backbone with training the detector from scratch (The full ablation can be referred to in Figure 13 of the attached PDF):

|  Variant | BLEU  | ROUGE-L | METEOR | Validity | Conformity | Freshness | Diversity |
| -------------------- | ----- | ------- | ------ | -------- | ---------- | --------- | --------- |
| w/o PreDet | 0.201 | 0.275   | 0.230  | 0.812    | 0.833      | 0.701     | 0.502     |
| Full Model           | 0.280 | 0.283   | 0.250  | 0.907    | 0.883      | 0.809     | 0.619     |

The ablation results show that omitting the pre-training step of the FasterRCNN model tends to result in the detection of more overlapping regions. This, in turn, causes a noticeable decrease in both knowledge diversity and freshness in the detected regions, which indicates the importance of loading the pre-trained model for region detection.

---

### Author Response · Authors · 2023-08-18
**Authors' message to all reviewers**

Dear reviewers,

We understand that chasing down your replies is not our job and we do not intend to add any pressure on your busy schedules. However, as we are getting closer to the end of the discussion phase, we would really appreciate it if you could be so kind to let us know if we have properly addressed some of your concerns in our rebuttal, and if anything can be further clarified.

Many thanks in advance!

Authors

---

### Decision · Program_Chairs · 2023-09-21

**Decision:**

Accept (poster)

**Comment:**

This paper introduces a new paradigm of open visual knowledge extraction called OpenVik. The proposed framework generates format-free knowledge by prompting a large multimodality model with detected regions of interest. It consists of an open relational region detector and a format-free visual knowledge generator. The paper highlights the limitations of existing approaches to visual knowledge extraction and demonstrates the correctness and uniqueness of the extracted open visual knowledge by OpenVik. The extracted knowledge is integrated across various visual reasoning applications, showing consistent improvements and indicating the real-world applicability of OpenVik.

Strengths:
1. Significance: open visual knowledge extraction is an attractive goal, which extracts knowledge from unlabeled image data.
2. Novelty: The proposed framework is a novel in the field.
3. Evaluation: solid evaluation of the generated knowledge, using traditional generative metrics and in-depth knowledge quality assessment. evaluation on various downstream tasks.
4. Analysis: the comparison of the extracted knowledge with non-parametric knowledge in existing KG and parametric knowledge from LLMs illustrates the unique value proposition of OpenVik.

Weaknesses:
1. The reviewers also have concerns over the complexity of the approach
2. Lack of ablation studies (e.g. knowledge-enhanced models, FasterRCNN).
3. The reviewers also like to see an important set of baselines involving multimodal models.